# Molecular basis and design principles of switchable front-rear polarity and directional migration in *Myxococcus xanthus*

Luís António Menezes Carreira[1], Dobromir Szadkowski[1], Stefano Lometto[2,3], Georg. K. A. Hochberg [2,3] & Lotte Søgaard-Andersen [1] ✉

During cell migration, front-rear polarity is spatiotemporally regulated; however, the underlying design of regulatory interactions varies. In rod-shaped *Myxococcus xanthus* cells, a spatial toggle switch dynamically regulates front-rear polarity. The polarity module establishes front-rear polarity by guaranteeing front pole-localization of the small GTPase MglA. Conversely, the Frz chemosensory system, by acting on the polarity module, causes polarity inversions. MglA localization depends on the RomR/RomX GEF and MglB/RomY GAP complexes that localize asymmetrically to the poles by unknown mechanisms. Here, we show that RomR and the MglB and MglC roadblock domain proteins generate a positive feedback by forming a RomR/MglC/MglB complex, thereby establishing the rear pole with high GAP activity that is non-permissive to MglA. MglA at the front engages in negative feedback that breaks the RomR/MglC/MglB positive feedback allosterically, thus ensuring low GAP activity at this pole. These findings unravel the design principles of a system for switchable front-rear polarity.

Cell polarity with the asymmetric localization of proteins within cellular space is ubiquitous and foundational for many cellular functions, including growth and motility[1–3]. Nevertheless, how polarity emerges at cellular scales from local protein-protein interactions and how it is dynamically controlled is poorly understood. Polarity regulators are often connected to generate networks that include positive feedback, negative feedback and/or mutual inhibition[2,4–7]. In transcriptional regulation, it is well-established that different designs of regulatory circuits can result in functionally equivalent outcomes, e.g. double-negative is functionally equivalent to double-positive regulation[8]. Similarly, polarity-regulating networks with functionally equivalent outcomes can have different designs, raising the question of why a particular network design has been selected.

A recurring theme in polarity-regulating systems is the localization of the active GTP-bound form of a small GTPase at a single intracellular location[6,7,9–12]. The GTPase, in turn, interacts with downstream effectors

to implement a specific response. These GTPases are molecular switches that alternate between an inactive, GDP-bound and an active, GTP-bound conformation[13]. The activation/deactivation cycle is regulated by a cognate guanine-nucleotide exchange factor (GEF), which facilitates the exchange of GDP for GTP, and a GTPase activating protein (GAP), which stimulates the low intrinsic GTPase activity[14]. Two experimentally and theoretically well-studied systems illustrate how polarity-regulating networks with different designs can result in equivalent outcomes. In *Saccharomyces cerevisiae* lacking the small GTPase Rsr1, the location of the single bud site depends on where the GTPase Cdc42 spontaneously forms a single cluster on the membrane. The responsible regulatory network centers on at least one positive feedback directly involving Cdc42[4,9]. Briefly, Cdc42-GTP spontaneously forms a cluster on the membrane and then recruits a complex that includes the GEF Cdc24[9]. Because Cdc24 activates additional Cdc42, Cdc24 recruitment stimulates the accumulation of additional

[1]Department of Ecophysiology, Max Planck Institute for Terrestrial Microbiology, 35043 Marburg, Germany. [2]Evolutionary Biochemistry Group, Max Planck Institute for Terrestrial Microbiology, 35043 Marburg, Germany. [3]Department of Chemistry and Center for Synthetic Microbiology, Philipps University, 35043 Marburg, Germany. ✉e-mail: sogaard@mpi-marburg.mpg.de

Cdc42-GTP, closing the positive feedback[9]. Cdc42 GAPs inhibit Cdc42 cluster growth and may be part of a negative feedback[9,15,16]. In the alternative system, unidirectional migration of the rod-shaped cells of the bacterium *Myxococcus xanthus* depends on the localization of the GTPase MglA at the leading front pole. In this case, the positive feedback does not involve MglA but rather the GAP MglB and the RomR scaffold[17]. Ultimately, these two proteins establish a rear, lagging pole with high GAP activity leaving only the opposite pole free to recruit MglA-GTP[17]. Thus, both systems generate a single Cdc42/MglA cluster. Here, we focus on the mechanistic basis of polarity establishment in *M. xanthus* and the functional properties conferred by the underlying network compared to the circuit that brings about Cdc42 cluster formation.

*M. xanthus* migrates unidirectionally on surfaces using two motility machines that assemble at the leading pole[11,18,19]. In response to signalling by the Frz chemosensory system, cells reverse the direction of movement[20]. During reversals, cells invert polarity and the pole at which the motility machines assemble switches[21,22]. Motility and its regulation by the Frz system are essential for multicellular morphogenesis with the formation of predatory colonies and spore-filled fruiting bodies[11,18,19]. Active MglA-GTP stimulates the assembly of the motility machineries at the leading cell pole[23–25]. Front-rear polarity is regulated dynamically by two interconnected protein modules, i.e. the polarity module and the Frz chemosensory system, that in combination generate a spatial toggle switch[19]. The polarity module sets up the leading/lagging polarity axis and, in addition to MglA, comprises four proteins that also localize asymmetrically to the cell poles (Fig. 1a). The homodimeric roadblock domain protein MglB alone has GAP activity and together with its low-affinity co-GAP RomY, forms the MglB/RomY complex with even higher GAP activity[26–28]. RomX alone has GEF activity and forms the RomR/RomX complex with even higher GEF activity and also serves as a polar recruitment factor for MglA-GTP[29].

Experiments and mathematical modelling have uncovered an intricate set of regulatory interactions between the proteins of the polarity module[17,26–32] (Fig. 1b). The RomR scaffold is at the base of all other polarity proteins' polar localization and also reinforces its own polar localization, thereby establishing a positive feedback[17]. RomR also engages in a positive feedback with MglB by an unknown mechanism[17]. Additionally, RomR directly recruits RomX to form the RomR/RomX GEF complex[29]. High concentrations of polar MglB stimulate polar recruitment of its low-affinity interaction partner RomY[28]. At the RomR node of the RomR/MglB positive feedback, RomR/RomX promotes MglA-GTP polar recruitment (Fig. 1b – connector from RomR/RomX to MglA)[29], and at the MglB node, MglB/RomY inhibits MglA-GTP polar recruitment (Fig. 1b – connector from MglB/RomY to MglA)[26–28]. Finally, MglA-GTP disrupts the RomR/MglB positive feedback by an unknown mechanism (Fig. 1b – connector from MglA to dashed box)[17]. Together these interactions have been suggested to result in the system's emergent properties (Fig. 1a, b)[17,28]. Briefly, at the pole with the highest RomR concentration, the RomR/MglB positive feedback establishes a pole with high concentrations of RomR/RomX and MglB/RomY. Due to the presence of the MglB/RomY complex, GAP activity dominates over GEF activity at this pole, thus inhibiting MglA-GTP recruitment, and this pole becomes the lagging pole. At the opposite pole, RomR/RomX GEF activity dominates over GAP activity because the low concentration of MglB is insufficient to recruit RomY[28]. Consequently, MglA-GTP is recruited to this pole and engages in the negative feedback to inhibit the RomR/MglB positive feedback, thereby ensuring the low concentration of the other polarity regulators. The Frz system is the second module of the spatial toggle switch, and the polarity module is the downstream target of this system. Frz signaling causes the inversion of polarity of the proteins of the polarity module by an unknown mechanism, thus laying the foundation for the assembly of the motility machineries at the new leading pole[30,31,33,34].

Among the interactions of the proteins of the polarity module, the positive feedback of RomR on itself, the RomR/MglB positive feedback, and the inhibitory effect of MglA-GTP on this positive feedback are poorly understood. MglC is also a homodimeric roadblock domain protein[35–37] and is involved in cell polarity regulation by an unknown mechanism[36]. Because MglC interacts with RomR and MglB[35,36], MglC was a candidate for acting in the RomR/MglB positive feedback.

Here, we show that MglC forms a complex with RomR and MglB, thereby establishing a RomR/MglC/MglB positive feedback and that MglA-GTP inhibits this positive feedback by breaking the interaction between the MglC and MglB roadblock domain proteins. Moreover, we demonstrate that the RomR/MglC/MglB positive feedback lays the foundation for switchable polarity.

## Results

### MglC is important for Frz-induced cellular reversals

To investigate the function of MglC in polarity, we recharacterized the motility defects of a mutant with an in-frame deletion of *mglC* (Δ*mglC*). In agreement with previous findings[36], the Δ*mglC* mutant has defects in both gliding and T4P-dependent motility in population-based motility assays, and ectopic expression of *mglC* complemented these defects (Supplementary Fig. 1a, b). In single cell-based motility assays (Supplementary Fig. 1c), and consistent with previous observations[36], Δ*mglC* cells moved with the same speed as wild-type (WT) for both motility systems; however, similarly to the Δ*frzE* negative control that lacks the FrzE kinase, Δ*mglC* cells had a significantly lower reversal frequency than WT.

To discriminate whether the Δ*mglC* mutant is unresponsive to or has reduced sensitivity to Frz signaling, we treated WT and Δ*mglC* cells with the short-chain alcohol isoamyl alcohol (IAA) that highly stimulates reversals in a FrzE-dependent manner[38]. WT and the Δ*mglC* mutant responded similarly to 0.3% IAA with the formation of colonies that had smooth edges and no visible flares on 0.5% agar, which is optimal for T4P-dependent motility, and few single cells at the edge on 1.5% agar, which is optimal for gliding motility (Supplementary Fig. 1a). Such smooth colony edges indicate a high reversal frequency[20,39]. We conclude that the Δ*mglC* mutant does not have a defect in motility per se but reduced sensitivity to Frz signalling resulting in a reduced reversal frequency.

### MglC is important for the polar localization of MglA, MglB and RomR

Because the polarity module is the downstream target of the Frz system, we quantified the polar localization of active, fluorescently labelled fusions of the polarity proteins in the absence of MglC. Because RomX localization follows that of RomR[29] and RomY localization follows the highest concentration of MglB[28], we used RomR and MglB localization as readouts for the localization of the RomR/RomX complex and MglB/RomY complex, respectively.

In snapshots of Δ*mglC* cells (Fig. 1c), polar localization of MglA-mVenus and MglB-mCherry was strongly reduced, while RomR-mCherry polar localization was only partially lost. MglA, MglB and RomR accumulated independently of MglC (Supplementary Fig. 1d).

### MglC polar localization depends partially on MglB and strongly on RomR

To study MglC localization, we first observed that a fully active MglC-mVenus fusion expressed from the native site (Supplementary Fig. 1a, b) localized in a bipolar asymmetric pattern with a large cluster at the lagging pole in WT cells and switched polarity during reversals (Fig. 1d). The bipolar asymmetric pattern was also evident in snapshots (Fig. 1e). In the absence of MglA, MglC-mVenus was more polar (Fig. 1e). However, in the absence of MglB, MglC-mVenus polar localization was partially lost; and, in the absence of RomR, it was almost completely lost (Fig. 1e). MglC-mVenus accumulated independently of

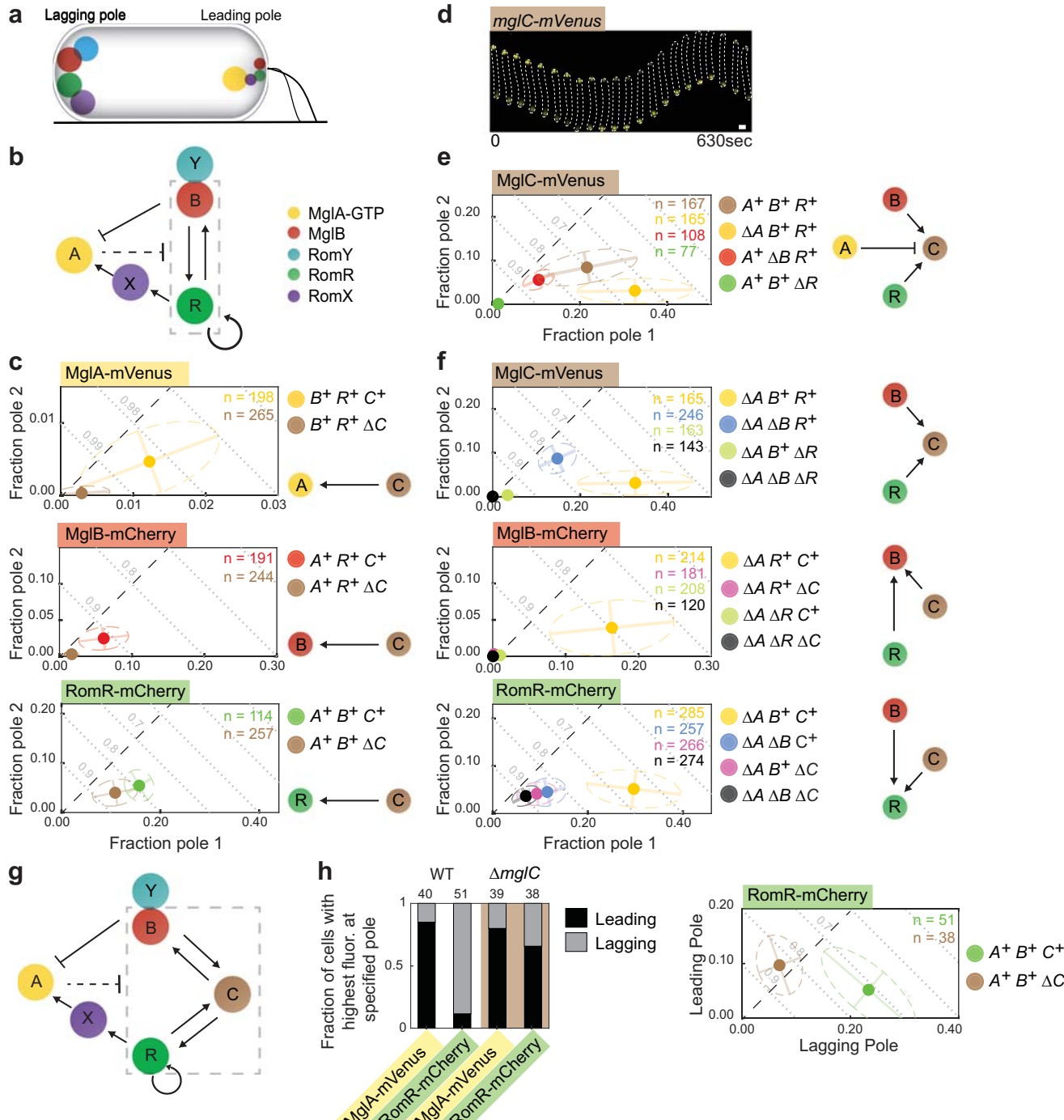

**Fig. 1 | MglC localization depends on MglA, MglB and RomR, and vice versa.**
**a** Schematic of the localization of the polarity proteins. T4P indicate the leading pole. The circle size indicates relative amount of a protein at a pole. Colour code as in Fig. 1b. **b** Schematic of interactions between polarity proteins. Dashed box indicates the RomR/MglB positive feedback. **c** MglA, MglB and RomR polar localization depends on MglC. All fusion proteins were synthesized from their native locus. In the diagrams, the poles with the highest and lowest polar fraction of fluorescence are defined as pole 1 and pole 2, respectively. Filled circles, mean fraction of fluorescence at each pole. Dispersion of the single-cell measurements is represented by error bars and ellipses (colored dashed lines). Black dashed lines, symmetry lines; grey dashed lines, guidelines indicating fraction of total polar fluorescence. Number of cells analyzed (n), indicated in top right corners. Similar results were obtained in three independent experiments; data from one experiment is shown. *mglA*, *mglB*, *mglC* and *romR* genotypes are indicated with *A*, *B*, *C* and *R*,

respectively. Schematics on the right, effects observed. **d** MglC-mVenus localizes asymmetrically and dynamically to the cell poles. The fusion was synthesized from the native locus. Cells were imaged at 30 sec intervals. 200 cells were imaged in two independent experiments; one representative cell is shown. Scale bar, 1μm. **e** MglC polar localization depends partially on MglB and strongly on RomR. Experiments were done and data presented as in Fig. 1c. **f** Quantification of the polar localization of MglC-mVenus, MglB-mCherry and RomR-mCherry in the absence of MglA. Experiments were done and data presented as in Fig. 1c. **g** MglC is a component of the RomR/MglC/MglB positive feedback (dashed box). **h.** MglC is essential for establishing correct RomR polarity. Cells were imaged as in Fig. 1d, and the fractions of cells with the brightest cluster at the leading or lagging pole determined. Left panel, summary of fractions of cells with indicated localization pattern. Total number of cells in three independent experiments is indicated on the top. Right panel, quantification of RomR-mCherry localization in moving cells.

MglA, MglB and RomR (Supplementary Fig. 1e). Thus, MglA inhibits MglC polar localization while MglC depends partially on MglB and strongly on RomR. Notably, in the absence of RomR, MglB fails to support significant MglC polar localization.

## MglC establishes the RomR/MglC/MglB positive feedback

Because the interpretation of the results for polar localization of MglC, MglB and RomR can be challenging due to the inhibitory effect of MglA-GTP on the RomR/MglB positive feedback, we quantified their polar fluorescence in strains lacking MglA.

MglC-mVenus polar localization in the $\Delta mglA\Delta mglB$ mutant was partially lost compared to the $\Delta mglA$ mutant, almost completely abolished in the $\Delta mglA\Delta romR$ mutant, and completely abolished in the triple $\Delta mglA\Delta mglB\Delta romR$ mutant (Fig. 1f). These observations confirm that MglC-mVenus polar localization depends partially on MglB and strongly on RomR. They also confirm that in the absence of RomR, MglB fails to support MglC polar localization significantly.

MglB-mCherry polar localization in the $\Delta mglA\Delta mglC$, the $\Delta mglA\Delta romR$ and the $\Delta mglA\Delta mglC\Delta romR$ mutants was almost completely lost (Fig. 1f). These results confirm that MglB-mCherry polar localization depends strongly on MglC and, as previously shown[17], on RomR. Moreover, neither MglC nor RomR alone can establish efficient polar MglB-mCherry localization.

RomR-mCherry polar localization was partially abolished in the $\Delta mglA\Delta mglB$, $\Delta mglA\Delta mglC$ and $\Delta mglA\Delta mglB\Delta mglC$ mutants (Fig. 1f). Thus, both MglB and MglC are important but not essential for RomR polar localization. Moreover, only when MglB and MglC are both present can they further stimulate RomR polar localization.

These observations demonstrate that RomR alone localizes polarly, and they support that RomR recruits MglC, which then recruits MglB. The observations that (1) MglB stimulates MglC polar localization in the presence of RomR, and (2) MglB together with MglC stimulates RomR polar localization support that the three proteins establish a positive feedback that reinforces their polar localization (Fig. 1g). These observations also suggest that the previously established RomR/MglB positive feedback depends on MglC, i.e. MglC helps to generate a RomR/MglC/MglB positive feedback by acting between RomR and MglB (Fig. 1g). Because MglA inhibits the RomR/MglB positive feedback[17], this model also explains the observation that MglA inhibits MglC polar localization (Fig. 1e). Moreover, the reduced MglA polar localization in the absence of MglC (Fig. 1c) is a direct outcome of the reduced RomR polar localization in the absence of MglC (Fig. 1c, f).

To further test the idea of the RomR/MglC/MglB positive feedback, we leveraged an established approach to monitor the cooperative polar recruitment of RomR-mCherry[17]. In this approach, a vanillate-inducible promoter drives *romR-mCherry* expression; upon induction, RomR-mCherry polar localization is followed by time-lapse fluorescence microscopy. To monitor RomR-mCherry synthesis over time, we estimate the RomR-mCherry concentration in individual cells, referred to as the fluorescence concentration, by measuring total cellular fluorescence and then normalizing by cell area, which we use as a proxy for cell volume.

Upon induction of *romR-mCherry* expression in the $\Delta mglA\Delta mglB\Delta romR\Delta mglC$ quadruple mutant (Supplementary Fig. 2a) and the $\Delta mglA\Delta romR\Delta mglC$ triple mutant (Supplementary Fig. 2b), RomR-mCherry localized asymmetrically to the poles at all fluorescence concentrations and quantitatively followed the pattern previously observed in the $\Delta mglA\Delta mglB\Delta romR$ triple mutant (Supplementary Fig. 2c). As described[17], the observations that the fractions of RomR-mCherry at both poles increase with fluorescence concentration at low induction levels provide evidence for positive cooperativity in RomR-mCherry polar localization (Supplementary Fig. 2a-d). Because RomR-mCherry polar localization is quantitatively similar in these three strains, we conclude that MglC, similar to MglB, is not essential for the positive feedback of RomR on itself. By contrast, in the $\Delta mglA\Delta romR$

double mutant, RomR-mCherry polar localization was increased and more asymmetric, with the brighter pole accounting for a larger fraction of RomR-mCherry fluorescence (Supplementary Fig. 2d). This observation confirms that MglC is essential for establishing the RomR/MglB positive feedback and that the RomR/MglB positive feedback is, in fact, a RomR/MglC/MglB positive feedback (Fig. 1g).

## MglC is essential for establishing correct RomR polarity

The model for polarity establishment (Fig. 1g) predicts that in the absence of MglC and, therefore, the RomR/MglC/MglB positive feedback, the residual polar RomR, together with RomX, will recruit MglA-GTP. As a result, MglA-GTP and RomR/RomX will have their highest polar fluorescence at the leading pole. To test this prediction, we performed time-lapse fluorescence microscopy of moving cells. In WT, MglA-mVenus localized with a large cluster at the leading pole and RomR-mCherry with a large cluster at the lagging pole in most cells (Fig. 1h). Importantly, and as predicted, in the $\Delta mglC$ mutant, MglA-mVenus and RomR-mCherry had their highest polar fluorescence at the leading pole in most cells (Fig. 1h). We conclude that MglC is important not only for the polar localization of MglA, MglB and RomR but also for establishing the correct polarity of RomR-mCherry.

## RomR, MglC and MglB interact to form a complex

To investigate the mechanism underlying the RomR/MglC/MglB positive feedback, we tested for direct interactions between RomR, MglC, MglB and MglA using pull-down experiments in vitro with purified proteins. In agreement with previous observations in in vitro pull-down experiments[35] and Bacterial Adenylate Cyclase-Based Two-Hybrid (BACTH) assays[36], Strep-MglC pulled-down His$_6$-MglB and MalE-RomR in pairwise combinations, but not MglA-His$_6$ preloaded with GTP (Fig. 2a; Supplementary Fig. 3a). In pairwise combinations using MalE-RomR as a bait, MalE-RomR pulled-down Strep-MglC but not His$_6$-MglB; notably, in the presence of all three proteins, RomR-MalE pulled down Strep-MglC as well as His$_6$-MglB (Fig. 2b; Supplementary Fig. 3b). Finally, in pairwise combinations, His$_6$-MglB pulled-down Strep-MglC but not MalE-RomR; however, in the presence of all three proteins, His$_6$-MglB pulled-down Strep-MglC as well as MalE-RomR (Fig. 2c; Supplementary Fig. 3c).

Next, we determined whether MglC and/or RomR/MglC have MglA GAP activity or interfere with MglB and/or MglB/RomY GAP activity. To this end, we determined MglA-His$_6$ GTPase activity in the presence of RomR, MglC, MglB and/or RomY. Neither Strep-MglC nor MalE-RomR/Strep-MglC affected MglA GTPase activity in the presence or absence of MglB-His$_6$ and/or Strep-RomY (Supplementary Fig. 4).

We conclude that RomR, MglC and MglB interact to form a complex in which MglC is sandwiched between RomR and MglB.

## The MglB KRK surface region represents the interface for interaction with MglC

To elucidate the structural basis for the RomR→MglC→MglB interactions, we took advantage of structural information for MglA, MglB and MglC[35,40–42]. Each MglB protomer in the homodimer consists of a five-stranded β-sheet sandwiched between the α2-helix and the α1/α3-helices[40]. In the dimer, the α2-helices generate the so-called two-helix side and the pairs of α1/α3-helices the so-called four-helix side (Fig. 3a). In the crystallographic structure of the MglA-GTPγS:MglB$_2$ complex, the MglA monomer interacts asymmetrically with the two-helix side of the MglB dimer[40–42].

Galicia et al. reported that the K14, R115, K120 residues in MglB are highly conserved in MglB homologs, surface-exposed in the solved structure of the MglB dimer generating two positively charged surface regions on the four-helix side of the MglB dimer[40] (Fig. 3a). Moreover, they reported that the MglB$^{K14A\ R115A\ K120A}$ variant (henceforth, MglB$^{KRK}$) with substitutions of these three positively charged residues to Ala still has GAP activity in vitro but localizes diffusely in vivo by an unknown

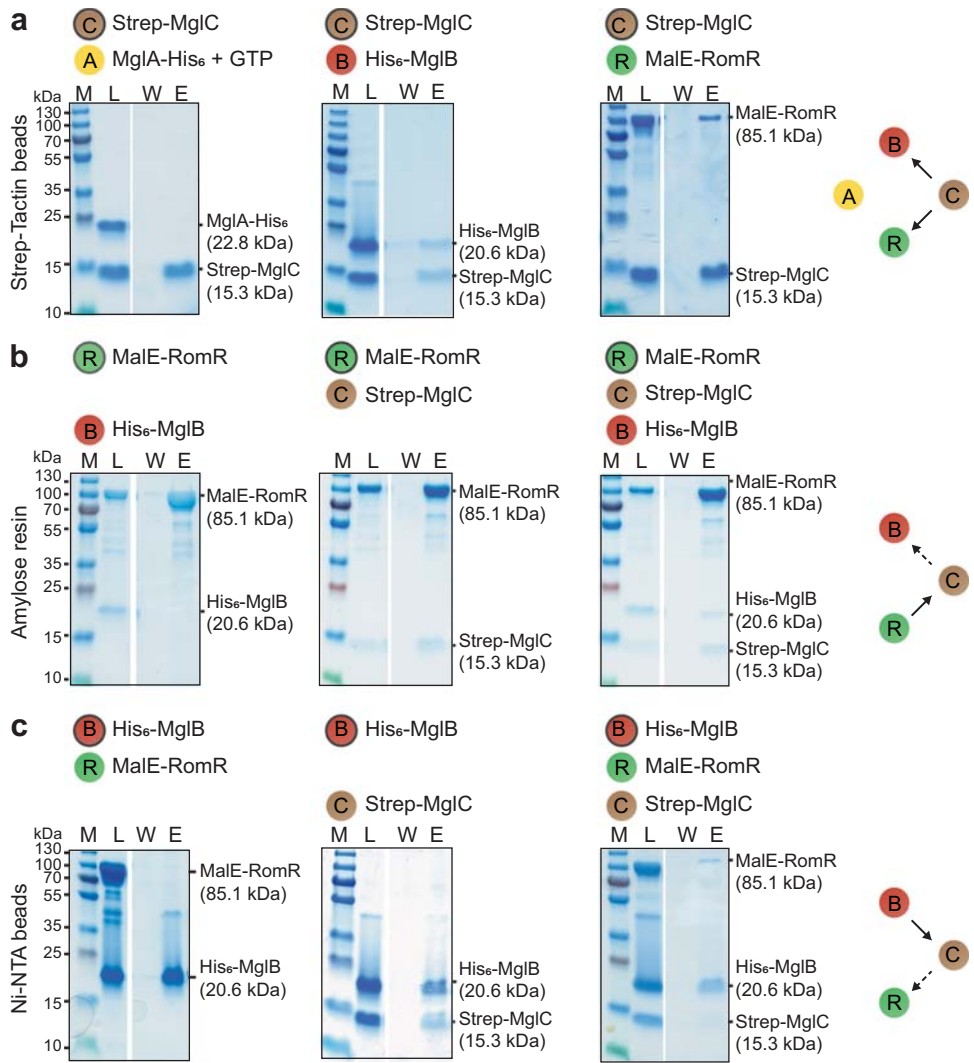

**Fig. 2 | MglB, MglC and RomR form a complex in vitro. a–c** Proteins were mixed at final concentrations of 10 μM and applied to the indicated matrices. Matrices were washed and bound proteins eluted. The bait protein is indicated by the black circle. In experiments with MglA-His$_6$, the protein was preloaded with GTP and all buffers contained 40 μM GTP. Equivalent volumes of the load (L), last wash (W) and eluate (E) were separated on the same SDS-PAGE gel and stained with Coomassie Brilliant Blue. Gap between lanes indicates lanes deleted for presentation purposes. Calculated molecular masses of the indicated proteins are indicated on the right and molecular weight markers (M) on the left. In the schematics on the right, a direct interaction with the bait is indicated by a black and an indirect interaction by a dashed arrow. For samples with MalE-RomR, the faster migrating proteins are degradation products. Similar results were obtained in three independent experiments.

mechanism[40]. Because MglB localizes diffusely in the absence of MglC, we, therefore, hypothesized that the positively charged surface regions in the MglB dimer defined by the K14, R115, K120 residues could be involved in the interaction between MglB and MglC.

In in vitro pull-down experiments, His$_6$-MglB$^{KRK}$ did not detectably bind Strep-MglC (Fig. 3b). Consistently, polar localization of MglB$^{KRK}$-mCherry in otherwise WT cells was strongly reduced independently of the presence or absence of MglC and MglA (Fig. 3c; Supplementary Fig. 5a). In the inverse experiment, MglB$^{KRK}$ caused a strong reduction in MglA-mVenus localization, while MglC-mVenus and RomR-mCherry polar localization was partially abolished (Fig. 3d; Supplementary Fig. 5a). We conclude that MglB$^{KRK}$ is deficient in interacting with MglC and suggest that the positively charged KRK surface regions in the MglB dimer represent the interface to MglC, in agreement with the suggestion by Kapoor et al.[35]. Moreover, we suggest that the effect of the MglB$^{KRK}$ variant on RomR and MglA localization is caused by the interruption of the RomR/MglC/MglB positive feedback, resulting in reduced polar RomR/RomX localization and, consequently, reduced MglA polar localization.

## The MglC FDI surface region represents the interface for interaction with MglB

The MglC homodimer's structure is similar to that of MglB with two-helix and four-helix sides (Fig. 4a)[35]. McLoon et al. reported that the F25, D26, I28 residues in MglC are highly conserved in MglC homologs[36] (Supplementary Fig. 6). In the solved structure of the MglC dimer, these three residues are surface-exposed and generate two separated negatively charged surface regions on the two-helix side (Fig. 4a). McLoon et al. also reported that the MglC$^{F25A\ D26A\ I28A}$ variant (henceforth, MglC$^{FDI}$) with substitutions of these three residues to Ala was abolished in its interaction with MglB but not with RomR based on BACTH assays[36]. We, therefore, hypothesized that the two negatively charged surface regions defined by the F25, D26, I28 residues could represent the interaction interface of MglC to MglB.

We initially sought to verify the effect of the MglC$^{FDI}$ variant on the MglC/MglB interaction in vitro; however, the Strep-tagged variant formed inclusion bodies in *Escherichia coli* under all conditions tested, precluding its purification. Importantly, in *M. xanthus*, MglC$^{FDI}$-mVenus was soluble (Supplementary Fig. 5b); however, it accumulated at a

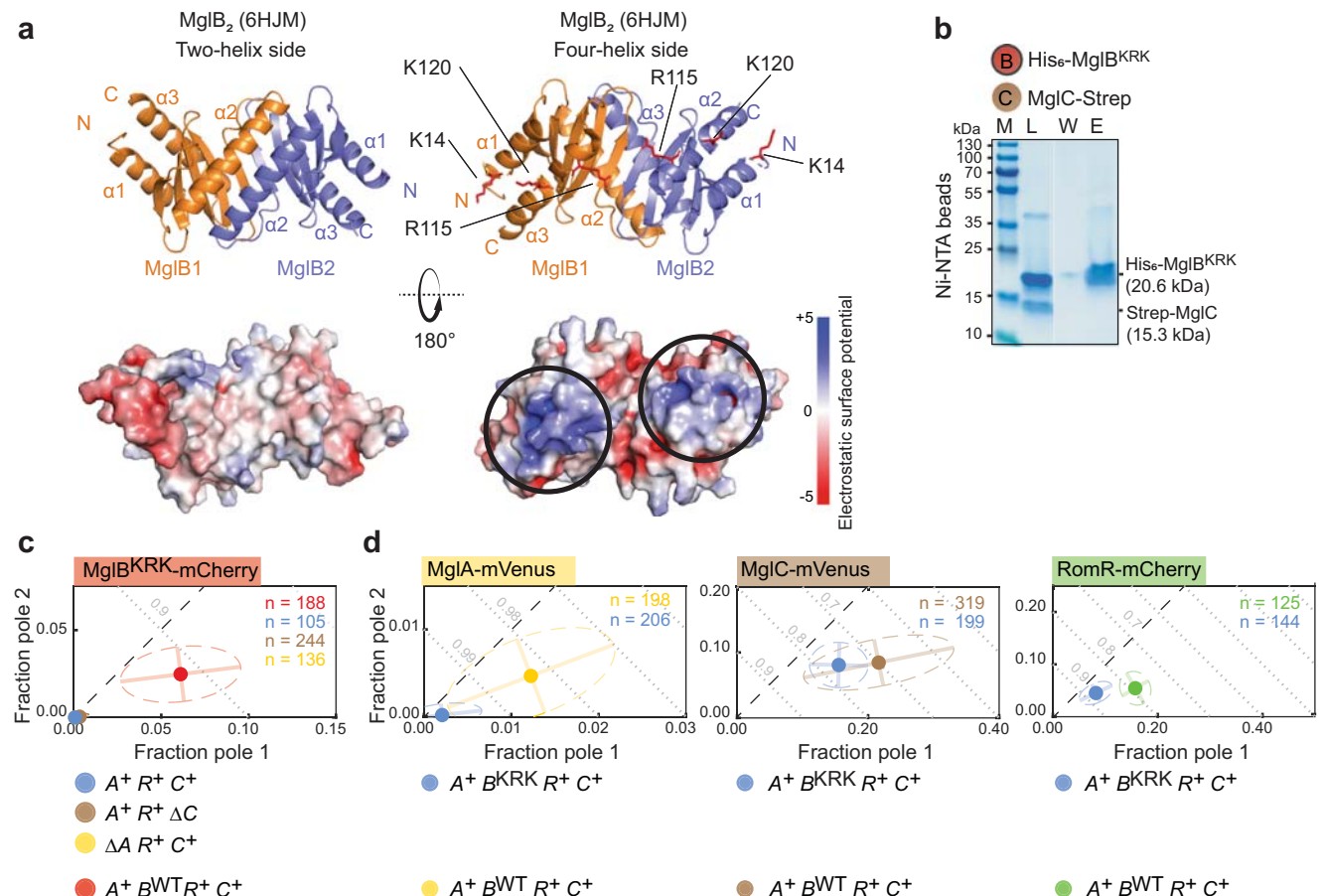

**Fig. 3 | The MglB KRK surface regions represent the interface for interaction with MglC. a** Crystallographic structure of MglB dimer (pdb ID: 6hjm[40]) viewed from the two-helix and four-helix sides. Lower panels, surface representation of MglB dimer based on electrostatic surface potential contoured from +5 to −5 kT e⁻¹. The K14, R115 and K120 residues are indicated in red on the four-helix side and the corresponding positively charged surface regions by black circles in the electrostatic surface potential diagrams. **b** The MglB^KRK variant does not interact with MglC. Pull-down experiment was performed with His₆-MglB^KRK as bait on the

indicated resin, and the data presented as in Fig. 2. Similar results were obtained in three independent experiments. **c.** MglB^KRK-mCherry has reduced polar localization. For comparison, MglB^WT-mCherry is included (red dot). MglB^KRK-mCherry was synthesized from the native locus. **d** MglB^KRK causes reduced polar localization of MglA, MglC and RomR. For comparison, the localization of the three fusion proteins are included in the presence of MglB^WT (yellow, brown and green dots). MglB^KRK was synthesized from the native locus. In **c**–**d**, experiments were done and data presented as in Fig. 1c.

reduced level compared to MglC-mVenus (Supplementary Fig. 5c). Polar localization of MglC^FDI-mVenus in otherwise WT cells was partially lost in comparison to MglC-mVenus (Fig. 4b). Moreover, MglC^FDI–mVenus polar localization did not change much upon removal of MglA or MglB but was abolished by removal of RomR (Fig. 4b). In the inverse experiment, MglC^FDI accumulated like MglC and, similar to the Δ*mglC* mutation, caused strong reductions in MglA-mVenus and MglB-mCherry polar localization while RomR-mCherry polar localization was only partially abolished (Fig. 4c; Supplementary Fig. 5c). We conclude that MglC^FDI is deficient in interacting with MglB but not with RomR and suggest that the negatively charged FDI surface regions in the MglC dimer represent the interface to MglB. Moreover, we suggest that the effect of the MglC^FDI variant on RomR and MglA localization is caused by the interruption of the RomR/MglC/MglB positive feedback, resulting in reduced polar RomR/RomX localization and, consequently, reduced MglA polar localization, as observed in the Δ*mglC* mutant.

**The MglC KRR surface region represents the interface for interaction with RomR**

In addition to the FDI residues, the K104, R106, R110 residues (MglC numbering) are highly conserved in MglC homologs (Supplementary Fig. 6). In the solved structure of the MglC dimer, these three residues are surface exposed on the four-helix side and define a continuous,

positively charged, surface-exposed region in the dimer (Fig. 4a). Because this region is apart from the FDI regions and MglC interacts with MglB and RomR in parallel, we hypothesized that the positively charged surface region defined by the K104, R106, R110 residues could represent the interaction interface of MglC to RomR.

To this end, we generated MglC^K104A R106A R110A variants (henceforth, MglC^KRR). However, the Strep-tagged variant formed inclusion bodies in *E. coli* under all conditions tested, precluding its purification. Importantly, in *M. xanthus*, MglC^KRR-mVenus was soluble (Supplementary Fig. 5b) and accumulated at the same level as MglC-mVenus (Supplementary Fig. 5d). Polar localization of MglC^KRR-mVenus in otherwise WT cells was strongly reduced compared to MglC-mVenus (Fig. 4d) and remained strongly reduced upon removal of MglA, MglB or RomR (Fig. 4d). In the inverse experiment, MglC^KRR accumulated like MglC and, similar to the Δ*mglC* mutation and MglC^FDI, caused strong reductions in MglA-mVenus and MglB-mCherry polar localization while RomR-mCherry polar localization was only partially abolished (Fig. 4e; Supplementary Fig. 5d). Based on these observations, we conclude that MglC^KRR is deficient in interacting with RomR and suggest that the two positively charged KRR surface regions in the MglC dimer represent the interface to RomR. Moreover, we suggest that the effect of the MglC^KRR variant on MglA localization is caused by the interruption of the RomR/MglC/MglB positive feedback, as observed in the Δ*mglC* and *mglC*^FDI mutants.

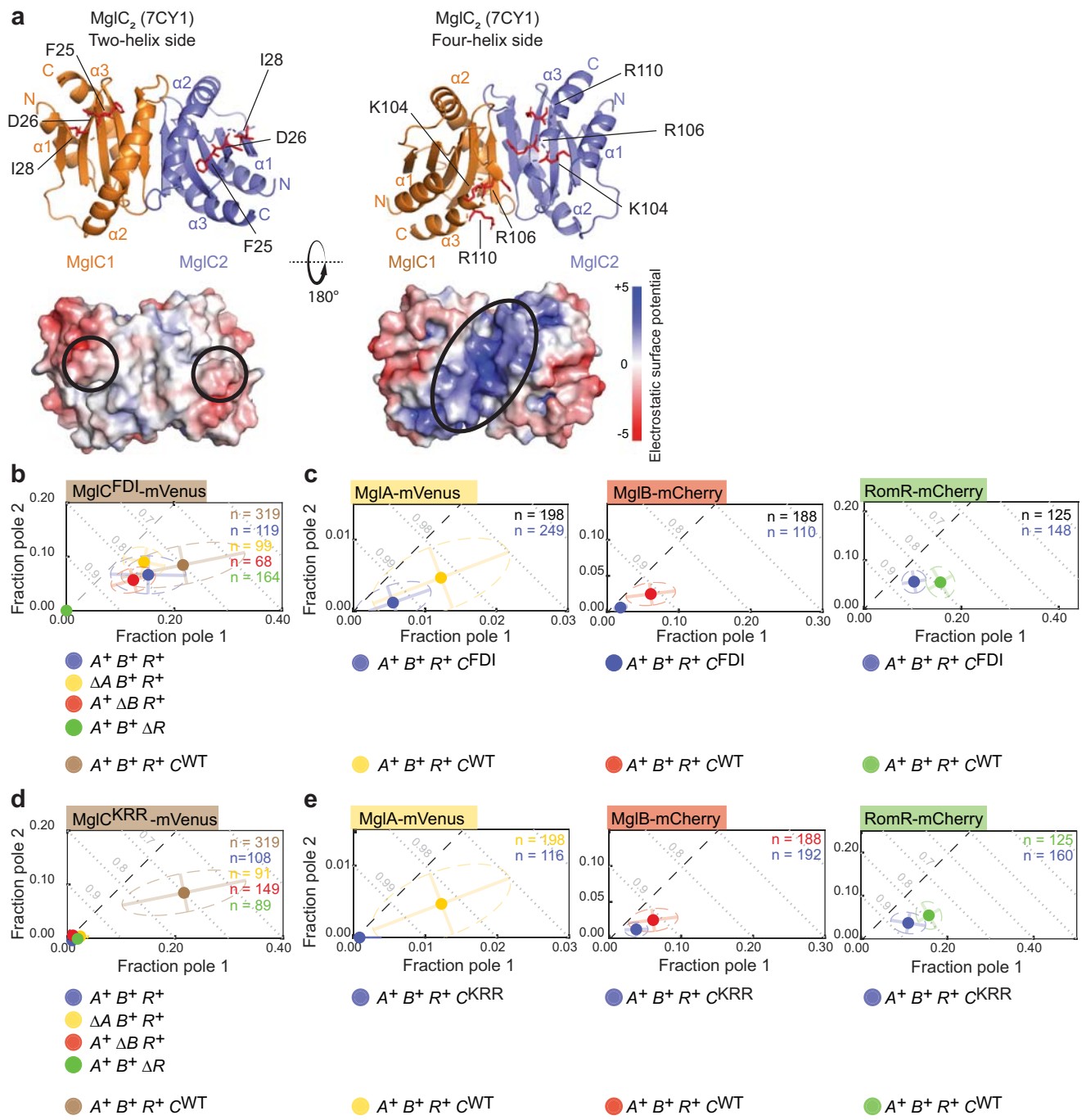

**Fig. 4 | The MglC FDI and KRR surface regions represent the interfaces for interaction with MglB and RomR, respectively. a** Crystallographic structure of MglC dimer (pdb ID: 7CY1[35]) viewed from two-helix and four-helix sides. Lower panels, surface representation of MglC dimer based on electrostatic surface potential contoured from +5 to −5 kT e⁻¹. The F25, D26, I28 residues and the K104, R106, R110 residues are indicated in red on the two-helix and the four-helix sides, respectively, and the corresponding negatively and positively charged surface regions indicated by black circles in the electrostatic surface potential diagrams. **b** MglC^FDI-mVenus has reduced polar localization. For comparison, MglC^WT-mVenus is included (brown dot). MglC^FDI-mVenus was synthesized from the native locus.

**c** MglC^FDI causes reduced polar localization of MglA, MglB and RomR. For comparison, the localization of the three fusion proteins is included in the presence of MglC^WT (yellow, red and green dots). MglC^FDI was synthesized ectopically. **d** MglC^KRR-mVenus has strongly reduced polar localization. For comparison, MglC^WT-mVenus is included (brown dot). MglC^KRR-mVenus was synthesized from the native locus. **e** MglC^KRR causes reduced polar localization of MglA, MglB and RomR. For comparison, the localization of the three fusion proteins is included in the presence of MglC^WT (yellow, red and green dots). MglC^KRR was synthesized ectopically. In **b-e**, experiments were done and data presented as in Fig. 1c.

## The α-helical RomR^C has three functions and represents the interface to MglC

RomR homologs comprise an N-terminal receiver domain of response regulators, an intrinsically disordered region (IDR), and an α-helical, negatively charged Glu-rich region at the C-terminus (henceforth,

RomR^C) (Fig. 5a)[30]. In BACTH assays, RomR^C interacts with MglC[36]. To examine whether RomR^C is the only interface to MglC, we generated a RomR^1−368 variant that lacks RomR^C and a variant only containing RomR^C.

First, using mass photometry (MP), we investigated the oligomeric structure of RomR. We detected MalE-RomR with masses matching well

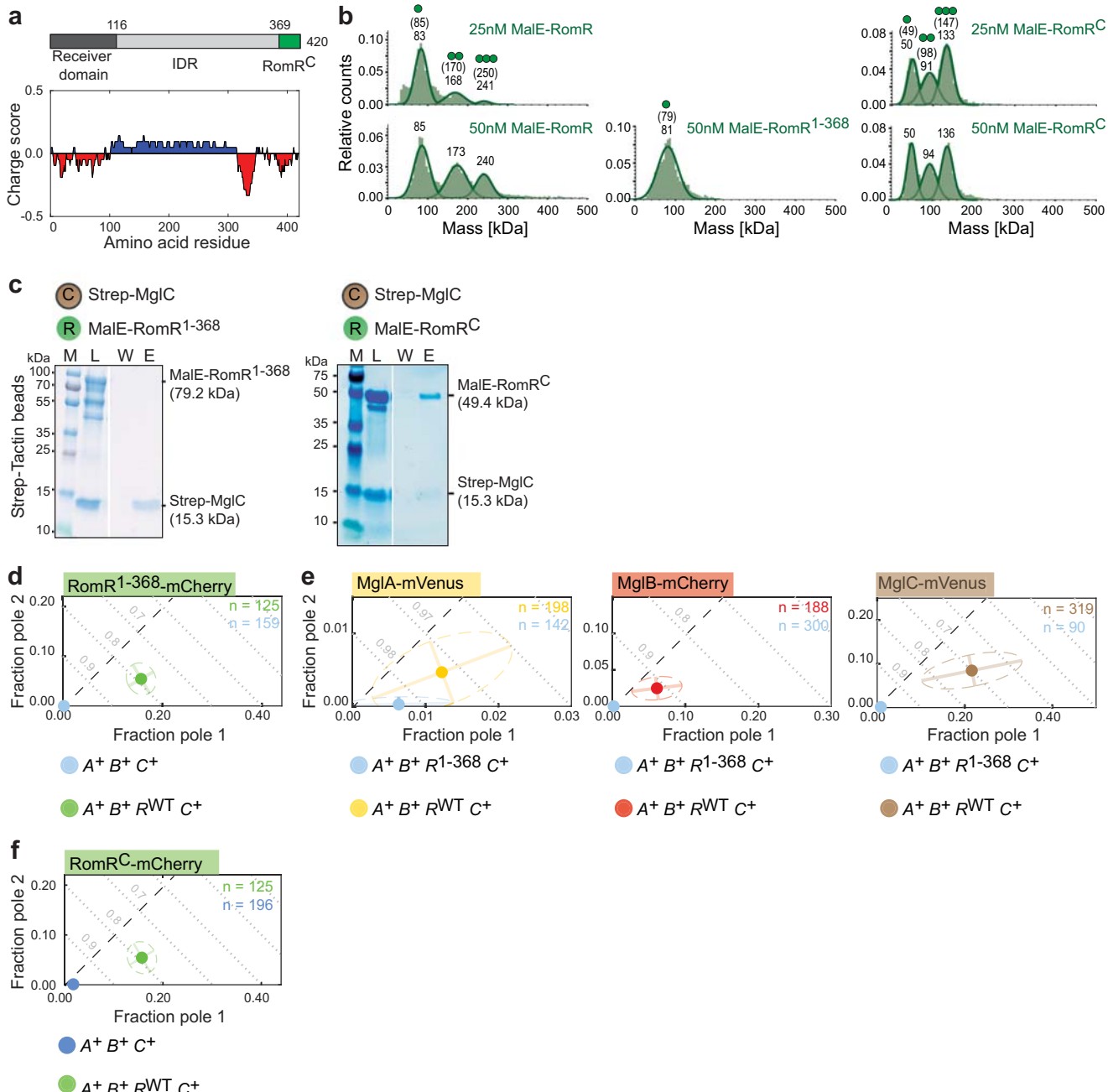

**Fig. 5 | RomR-C has three functions and represents the interface for interaction with MglC. a** Domain architecture and charge score of RomR. Charge score was calculated using a sliding window of 20 residues. **b** MP analysis of MalE-RomR, MalE-RomR$^{1-368}$ and MalE-RomR$^C$. Molecular masses corresponding to the respective Gaussian fits are shown in kDa above the fitted curves. Calculated molecular masses of the monomeric, dimeric and trimeric MalE-RomR variants are indicated in brackets together with symbols of oligomeric states. Similar results were obtained in three independent experiments. **c** RomR$^C$ interacts with MglC while RomR$^{1-368}$ does not. Pull-down experiment was performed with Strep-MglC as bait on the indicated resin and presented as in Fig. 2. The multiple bands below MalE-

RomR$^C$ and MalE-RomR$^{1-368}$ are degradation products. Similar results were obtained in three independent experiments. **d** RomR$^{1-368}$-mCherry has strongly reduced polar localization. For comparison, RomR$^{WT}$-mCherry is included (green dot). RomR$^{1-368}$-mCherry was synthesized from the native locus. **e** RomR$^{1-368}$ causes strongly reduces polar localization of MglA, MglB and MglC. For comparison, the localization of the three fusion proteins is included in the presence of RomR$^{WT}$ (yellow, red and brown dots). RomR$^{1-368}$ was synthesized from the native locus. **f** RomR$^C$-mCherry localizes polarly. For comparison, RomR$^{WT}$-mCherry is included (green dot). RomR$^C$-mCherry was synthesized ectopically. In d-f, experiments were done and data presented as in Fig. 1c.

with monomers, dimers and trimers (Fig. 5b). MalE-RomR$^C$ was also detected with masses matching well monomers, dimers and trimers, while MalE-RomR$^{1-368}$ was only detected at a mass matching monomers (Fig. 5b). Trimeric MalE-RomR was more prevalent at 50 nM compared to 25 nM, while trimeric MalE-RomR$^C$ was equally present at 25 nM and 50 nM (Fig. 5b). We conclude that RomR$^C$ is required and sufficient for RomR oligomerization and that the receiver domains and the

IDRs do not interact. Moreover, these results support that RomR forms up to trimers and full-length RomR begins to dissociate to dimers below 50 nM. Based on quantitative immunoblot analysis, an *M. xanthus* cell contains ~6000 ± 2000 RomR molecules (Supplementary Fig. 5e), resulting in a cellular RomR concentration of ~2.5 ± 0.8 μM. We, therefore, suggest that RomR is predominantly present as a trimer in vivo.

In pull-down experiments, MalE-RomR[1-368] did not detectably interact with Strep-MglC while MalE-RomR[C] did (Fig. 5c). In *M. xanthus*, RomR[1-368]-mCherry accumulated at the same level as RomR-mCherry (Supplementary Fig. 5f), however polar localization in otherwise WT cells was abolished (Fig. 5d). In the inverse experiment, RomR[1-368] accumulated like RomR and, similar to the Δ*romR* mutation, caused strong reductions in the polar localization of MglA-mVenus, MglB-mCherry and MglC-mVenus (Fig. 5e[17,30,31]; Supplementary Fig. 5f). RomR[C]-mCherry, even when expressed from the strong *pilA* promoter, accumulated at a strongly reduced level (Supplementary Fig. 5f); importantly, most cells had a weak polar signal (Fig. 5f).

We conclude that the negatively charged RomR[C] has three functions: It is required and sufficient for RomR oligomerization, represents the RomR interface to MglC, and is required and at least partly responsible for the polar localization of RomR.

## A structural model of the RomR/MglC/MglB complex

To gain structural insights into the RomR/MglC/MglB complex, we used structural information[35,40,41], our functional data and AlphaFold-Multimer[43-45] structural predictions to model this complex. The AlphaFold-Multimer models of the MglB and MglC dimers were predicted with high confidence and agreed well with the solved, crystallographic structures of the MglB and MglC dimers (Supplementary Fig. 7a-c), documenting the validity of the structural predictions.

A low-resolution structure of the MglC/MglB complex supports that one MglC dimer binds two MglB dimers[35]. In AlphaFold-Multimer models with the same stoichiometry, two MglB dimers are predicted with high accuracy to interact using their four-helix sides with the "lateral" edges of the two-helix side of the MglC dimer giving rise to an $MglC_2:(MglB_2)_2$ complex (Fig. 6a; Supplementary Fig. 7d). In this complex, Pymol-based analyses support that the R115 residues in the two KRK regions of an MglB dimer are in close proximity to and involved in establishing contact with D26 and I28 of an MglC FDI region (Fig. 6a, inset). Thus, this structural model agrees with a 2:4 stoichiometry of the MglC/MglB complex and together with our experimental findings support that the oppositely charged MglB KRK and MglC FDI surface regions interface. We note that the MglB dimers bound to MglC are structurally different compared to the MglB dimer in isolation, i.e. the four-helix side is in a more closed state when complexed with $MglC_2$ (Supplementary Fig. 7e). On the other hand, the MglC dimer bound to MglB only has minor structural differences compared to the dimer in isolation (Supplementary Fig. 7f).

To determine the stoichiometry of the RomR/MglC complex, we used MP. We detected a MalE-MglC fusion protein with masses matching a monomer and dimer (Fig. 6b). In the presence of both MalE-MglC and MalE-RomR, we detected, in addition to the masses of the individual proteins, complexes with masses consistent with a RomR:MglC stoichiometry of 2:2 and 3:2 (Fig. 6b; see also Fig. 5b). Similarly, in the presence of both MalE-MglC and MalE-RomR[C], we detected additional peaks with masses consistent with a RomR[C]:MglC stoichiometry of 2:2 and 3:2 (Fig. 6b; see also Fig. 5b).

To obtain structural insights into the RomR/MglC complexes, we attempted to generate AlphaFold-Multimer structural models of dimeric and trimeric RomR, dimeric and trimeric RomR[C], $RomR_2:MglC_2$ and $RomR_3:MglC_2$, as well as $RomR^C_2:MglC_2$ and $RomR^C_3:MglC_2$ complexes. However, none of these complexes was predicted with high confidence. Altogether, our experimental data support that the MglC dimer can interact with dimeric and trimeric RomR and that the interface between MglC and RomR are represented by the oppositely charged KRR regions in the MglC dimer and the negatively charged RomR[C] in the RomR dimer and trimer.

In total, these data support that a single MglC dimer is sandwiched between two MglB dimers and a RomR dimer or trimer, giving rise to a RomR:MglC:MglB complex with a 2:2:4 or a 3:2:4 stoichiometry. Because quantitative immunoblot analysis (Supplementary Fig. 5e)

support that RomR is predominantly present as a trimer in vivo, we suggest that the dominant form of the RomR:MglC:MglB complex in vivo has a 3:2:4 stoichiometry (Fig. 6c).

## MglC and MglB decrease RomR-mCherry polar turnover

The structural model of the RomR/MglC/MglB complex support a model for how RomR, MglC and MglB interact and how polar RomR recruits MglC, which recruits MglB. However, from this model, it is not clear how the positive RomR/MglC/MglB feedback is closed. We speculated that this loop could be closed if RomR would bind more stably to the poles in the RomR/MglC/MglB complex compared to RomR alone. To obtain a metric for the stability of RomR in polar clusters, we used Fluorescence Recovery after Photobleaching (FRAP) experiments in which polar RomR-mCherry clusters were bleached and half-maximal recovery time ($T_{1/2}$) and the mobile fraction ($F_{mob}$) used to assess RomR-mCherry turnover. In WT, RomR-mCherry at the lagging/leading pole dynamically exchanged with the cytoplasm with $T_{1/2}$ of $25.7 \pm 15.2/17.3 \pm 8.6$ s, similar to previous results[34], and $F_{mob}$ of $0.7 \pm 0.1/0.9 \pm 0.1$ (Fig. 6d, e). Thus, RomR-mCherry turnover is significantly slower and less at the lagging than at the leading pole. These observations agree with MglA-GTP at the leading pole engaging in a negative feedback to inhibit the RomR/MglC/MglB positive feedback. Consistently, in the non-motile Δ*mglA* mutant in which leading and lagging poles cannot be distinguished, $T_{1/2}$ was increased and $F_{mob}$ decreased compared to the leading pole in WT (Fig. 6e). Importantly, in Δ*mglB* and Δ*mglC* cells, $T_{1/2}$ and $F_{mob}$ of RomR-mCherry at the two poles were similar, and the $T_{1/2}$ values significantly lower and the $F_{mob}$ values significantly higher than at the lagging pole in WT (Fig. 6e).

These observations support that MglB and MglC jointly reduce polar RomR-mCherry turnover, thus supporting that more stable polar binding of RomR in the presence of both MglC and MglB closes the RomR/MglC/MglB positive feedback.

## MglA-GTP breaks the MglC/MglB interaction

To dissect how MglA-GTP inhibits the positive RomR/MglC/MglB feedback, we hypothesized that MglA-GTP breaks the interaction between RomR/MglC, MglC/MglB or both. To this end, we performed pull-down experiments with Strep-MglC as bait. Strep-MglC pulled-down $His_6$-MglB and MalE-RomR but not MglA-$His_6$ in the presence of either the non-hydrolyzable GTP analogue GppNHp or GDP (Fig. 7a, b; Supplementary Fig. 8a). Intriguingly, in the presence of MglA-$His_6$ loaded with GppNHp, Strep-MglC no longer pulled-down $His_6$-MglB but still pulled-down $His_6$-MglB in the presence of MglA-$His_6$ loaded with GDP (Fig. 7a). By contrast, Strep-MglC pulled-down MalE-RomR in the presence of MglA-$His_6$ loaded with either GppNHp or GDP (Fig. 7b). We conclude that MglA-GTP specifically breaks the MglC/MglB interaction but not the MglC/RomR interaction. This observation is also in agreement with neither MglC nor RomR/MglC affecting MglB and MglB/RomY GAP activity (Supplementary Fig. 4). Thus, addition of MglA-GTP to the RomR/MglC/MglB complex results in sequestration of MglB by MglA-GTP, thus allowing MglB GAP activity to proceed.

To understand the basis for MglA-GTP inhibition of the MglC/MglB interaction, we compared the solved structure of MglA-$GTP_\gamma S$:$MglB_2$ to the $MglC_2:(MglB_2)_2$ AlphaFold-Multimer model (Fig. 6a). We identified significant conformational differences in the MglB dimers in the two complexes. Specifically, the $MglB_2$ four-helix side is in a more open state when complexed with MglA-$GTP_\gamma S$ than with $MglC_2$ (Fig. 7c). As a result, the two R115 residues in the MglB KRK regions are likely positioned in such a way in the complex with MglA-$GTP_\gamma S$ that they cannot interact with D26 and I28 in the MglC FDI region (Fig. 7c, d; see also Fig. 6a). We note that in the solved structure of the MglB dimer with MglA-$GTP_\gamma S$, the four-helix side is more open than in the solved structure of the MglB dimer in isolation (Supplementary Fig. 8b). Thus, these solved structures together with the

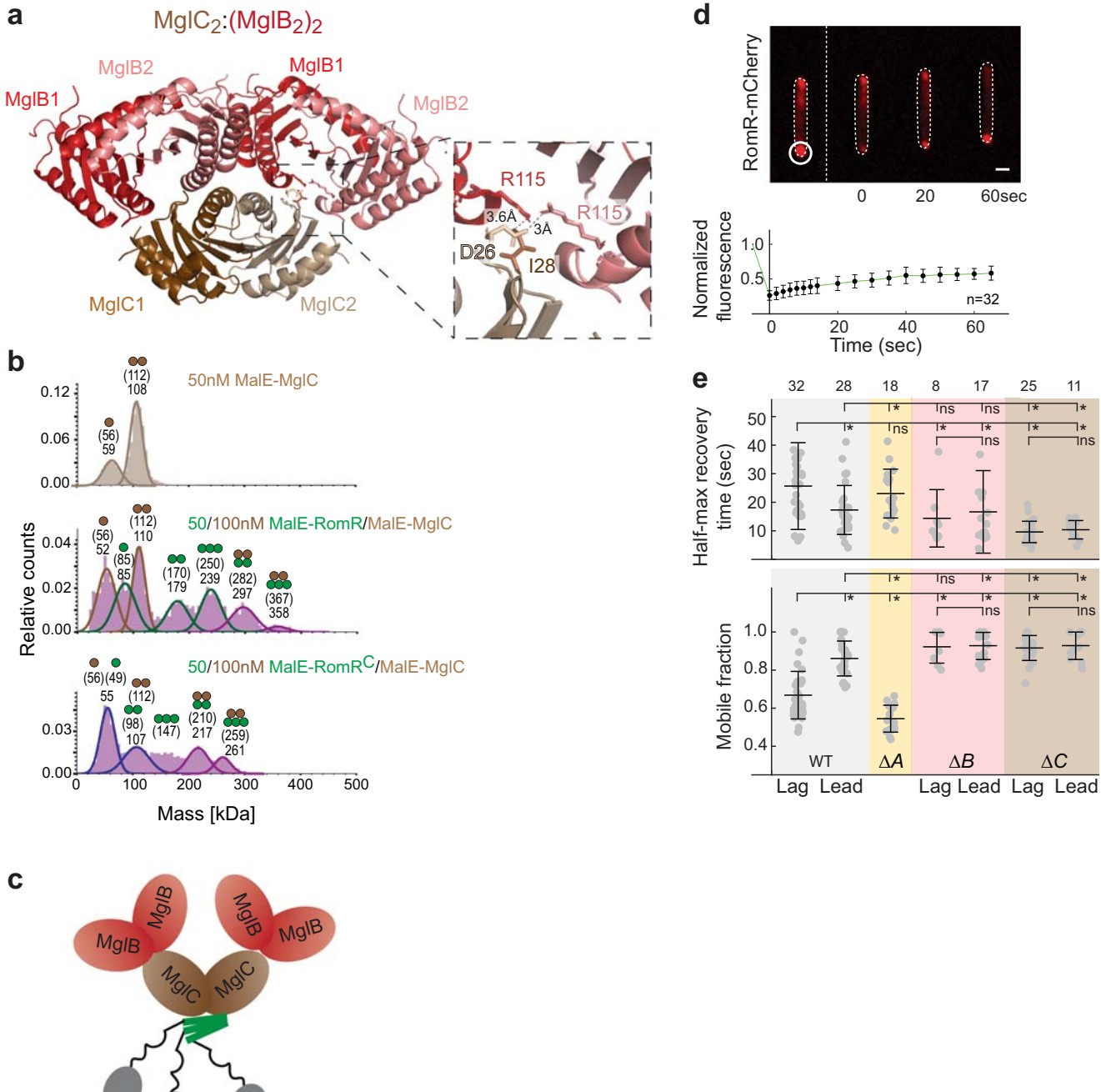

**Fig. 6 | MglC and MglB stabilize polar RomR-mCherry binding. a** AlphaFold-Multimer structural model of MglC$_2$:(MglB$_2$)$_2$ complex. Model rank 1 is shown. Inset shows R115 in each of the MglB protomers together with the D26 and I28 residues in a MglC protomer. **b** MP analysis of MalE-MglC (top) and mixtures of MglC with MalE-RomR or MalE-RomR$^C$. Molecular masses corresponding to the respective Gaussian fits are shown in kDa above the fitted curves. The Gaussian fits are colored according to the respective protein MalE-MglC (brown), MalE-RomR (green), MalE-RomR$^C$ (green), MalE-RomR:MalE-MglC (purple) and MalE-RomR$^C$:MalE-MglC (purple). Blue Gaussian fit (bottom panel) indicates a mixture of MalE-MglC and MalE-RomR$^C$. Calculated molecular masses of monomeric and dimeric MalE-MglC, monomeric, dimeric and trimeric MalE-RomR/MalE-RomR$^C$, and MalE-RomR/MalE-RomR$^C$:MalE-MglC complexes with stoichiometries of 2:2 and 3:2 are indicated in brackets together with symbols of the oligomeric states. Similar results were

obtained in three independent experiments. **c** Schematic of the RomR/MglC/MglB complex with a 3:2:4 stoichiometry. **d** Measurement of in vivo recovery kinetics of polar RomR-mCherry clusters in FRAP experiments. Upper panel, white circle indicates the bleached region of interest (ROI) at a lagging pole and the stippled line the bleaching event. Lower panel, normalized fluorescence intensity of the ROI before bleaching was set to 1.0. Colored dots indicate the mean and error bars STDEV. Dark lines show the recovery fitted to a single exponential. n, number of bleaching events at a lagging pole. Scale bar, 2μm. **e** Summary of T$_{1/2}$ and F$_{mob}$. Cells were treated as in d with bleaching of clusters at the lagging (Lag) or leading (Lead) pole. Total number of bleaching events in three biological replicates are listed above. Error bars, mean ± STDEV. *$P < 0.05$, ns, no significant difference two-sided Student's t-test. The exact $P$ values are listed in the Source Data File.

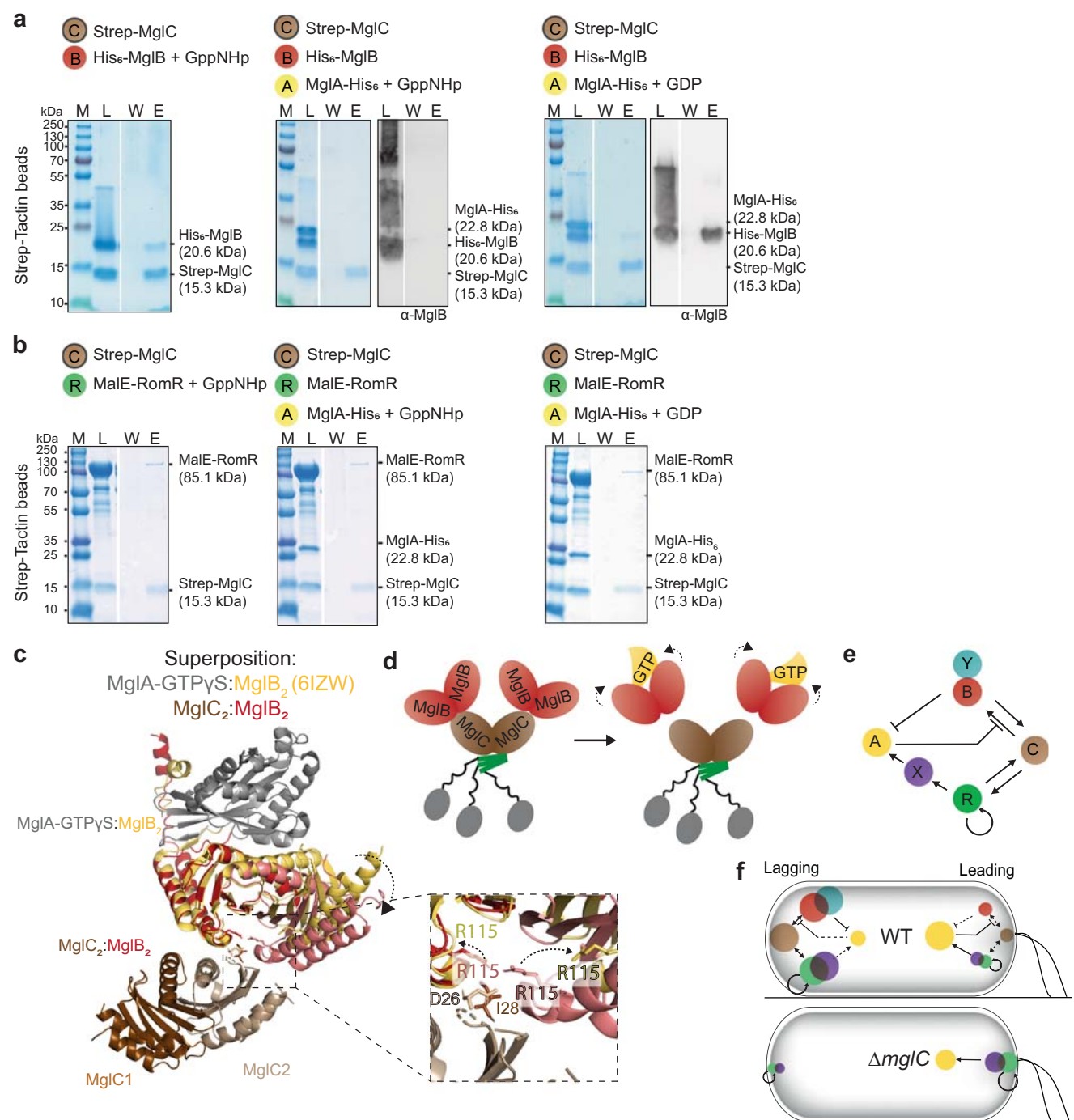

**Fig. 7 | MglA-GTP breaks the interaction between MglC and MglB. a, b** MglA-GTP breaks the interaction between MglC and MglB. Pull-down experiments were performed with Strep-MglC as bait on the indicated resin as described in Fig. 2. MglA-His₆ was preincubated with GppNHp or GDP (final concentration 40 μM). All buffers contained 40 μM GppNHp or GDP. In **a**, the SDS-PAGE gels were probed by immunoblotting with α-MglB antibodies. In b, the multiple bands below MalE-RomR are degradation products. Similar results were obtained in three independent experiments. **c.** Crystallographic structure of MglA-GTPγS:MglB₂ (grey/yellow) (pdb ID: 6izw[41]) superimposed on the AlphaFold-Multimer model of a MglC₂:MglB₂ complex (brown/red). Inset, R115 in each of the MglB protomers in the MglA-GTPγS:MglB₂ complex is shown in yellow and in red in the MglC₂:MglB₂

model; D26 and I28 are shown in brown for one of the MglC protomers. Arrows indicate the repositioning of R115 in the two complexes. For simplicity, the MglC dimer is shown to interact with only one MglB dimer. **d** Schematic of the breaking of the MglB/MglC interaction by MglA-GTP. Bent arrows indicate the conformational change in the MglB dimers upon binding of MglA-GTP. **e** Regulatory interactions that establish and maintain front-rear polarity in *M. xanthus*. **f** Different interactions between the polarity proteins dominate at the leading and lagging poles in WT and the Δ*mglC* mutant. Full arrows show locally strong interactions, dashed arrows show interactions that are locally suppressed. The arrow from RomR on itself indicates the positive feedback that reinforces its polar localization but it is not known how RomR becomes polarly localized. Color code as in **e**.

AlphaFold-Multimer model of the MglC₂:(MglB₂)₂ complex support that the MglB dimer can exist in three different conformational states, where the degree of "openness" on the four-helix side varies (Supplementary Fig. 8c).

## Discussion

Here, we identify MglC as a critical component of the polarity module for switchable front-rear polarity in *M. xanthus*. We demonstrate that the previously proposed RomR/MglB positive feedback incorporates

and depends on MglC. These three proteins form a heteromeric RomR/MglC/MglB complex in which MglC is sandwiched between RomR and MglB. In vivo, they establish the RomR/MglC/MglB positive feedback that results in the colocalization of the RomR/RomX GEF and MglB/RomY GAP at high concentrations at the lagging pole (Fig. 7e and f, upper panel). Moreover, we demonstrate that the previously reported inhibitory effect of MglA-GTP on the RomR/MglB positive feedback is the result of MglA-GTP breaking the MglC/MglB interaction without interfering with the RomR/MglC interaction in the RomR/MglC/MglB positive feedback (Fig. 7e and f, upper panel). By way of this inhibitory effect, MglA-GTP at the leading pole limits the accumulation of the other polarity regulators at this pole. By engaging in these interactions, MglC stimulates polar localization of the remaining polarity proteins and is also key to enabling dynamic inversion of polarity in response to Frz signaling.

In vitro observations together with an AlphaFold-Multimer structural model of the MglC/MglB complex and in vivo experiments, support that the RomR:MglC:MglB complex has a 3:2:4 stoichiometry in vivo. Specifically, our data support that the negatively charged α-helical RomR[C] interacts with the two juxtaposed positively charged KRR surface regions in the MglC dimer, and that each of the two negatively charged FDI surface regions in the MglC dimer interface with the positively charged KRK surface regions in a MglB dimer. These interactions between oppositely charged surface regions allow polar RomR to recruit MglC, which recruits MglB. FRAP experiments in vivo demonstrated that MglC and MglB enable more stable polar RomR occupancy. Based on these findings, we infer that the RomR/MglC/MglB positive feedback for polar localization involves direct recruitment via the RomR→MglC→MglB interactions. These interactions stabilize polar RomR binding, thereby closing the positive feedback. Because neither RomR, MglC, nor RomR/MglC has measurable GAP activity or measurably affects GAP activity by MglB/RomY and MglB, we infer that one function of MglC is to connect MglB and RomR to establish the positive feedback.

In vitro, MglA-GTP breaks the MglC/MglB interaction in the RomR/MglC/MglB complex without interfering with the RomR/MglC interaction. A comparison of the solved structure of the MglA-GTPγS:MglB$_2$ complex with an AlphaFold-Multimer model of the MglC$_2$:(MglB$_2$)$_2$ complex supports that MglA-GTP breaks the MglC/MglB interaction using an allosteric mechanism. In this model, MglA-GTP by binding to the two-helix side of a MglB dimer induces a conformational change that alters the four-helix side of the MglB dimer, thereby breaking the interaction between the MglC FDI and MglB KRK interfaces. Supporting these observations, RomR/MglC neither affects MglB nor MglB/RomY GAP activity (Supplementary Fig. 4). Thus, the second function of MglC is to enable the inhibitory effect of MglA-GTP on the RomR/MglC/MglB positive feedback in vivo. In our model, the high concentration of the MglB/RomY GAP complex at the lagging pole inhibits MglA-GTP recruitment to this pole by stimulating MglA GTPase activity. In this process, MglA-GTP breaks the RomR/MglC/MglB positive feedback resulting in the detachment of MglB. However, due to the high concentration of RomR and MglC at this pole, MglB will rapidly be recaptured to restore the RomR/MglC/MglB complex. Consistent with this notion, polar MglB exchanges rapidly with the cytoplasm with a $T_{1/2}$ of ~6 sec in FRAP experiments[34].

Our study raises several intriguing questions for future research regarding the proteins of the polarity module. First, RomR[C] has three functions: It interacts not only with MglC but also mediates oligomerization with dimer and trimer formation and is required and at least partly responsible for RomR polar localization. In vivo quantification of the RomR concentration (Supplementary Fig. 5e) suggests that trimeric RomR is the predominant form in vivo; however, which of the two forms represent the active form remains to be determined. Similarly, it is not known how RomR[C] brings about the polar localization of RomR, and how RomR stimulates its polar binding. Second,

experimental evidence and AlphaFold-Multimer structural models support that MglB binds its co-GAP RomY with low affinity on the two-helix side[28]. We, therefore, suggest that the RomR/MglC/MglB complex at the lagging pole also contains RomY forming a RomR/MglC/MglB/RomY complex. Third, RomR interacts with RomX to generate the polarly localized Rom/RomX GEF complex. While this complex's structural details are unknown, they raise the possibility that the RomR/MglC/MglB complex may also include RomX. The complexes formed will be addressed in future work.

The Δ*mglC* mutant resembles WT concerning unidirectional motility but is less sensitive to Frz signaling, supporting that the ultimate function of MglC is to establish sensitivity to Frz signaling, thereby enabling polarity inversions. The two output response regulators of the Frz system, FrzX and FrzZ, act on the polarity module by unknown mechanisms to enable polarity inversions[30,31,33,34]. The observation that the Δ*mglC* mutant still responds to high levels of Frz signaling argues that MglC is not the downstream molecular target of the Frz system but enables Frz responsiveness by a different mechanism. As predicted by the model for polarity establishment (Fig. 7e, f, upper panels), neither MglC nor the RomR/MglC/MglB positive feedback is important for MglA localization at the leading pole. Instead, in the absence of MglC, and therefore, the RomR/MglC/MglB positive feedback, the highest polar concentration of the RomR/RomX complex colocalizes with MglA at the leading pole (Fig. 7f, lower panel). In the Δ*mglC* mutant, RomR/RomX and MglA polar localization is driven by RomR stimulating its own polar binding in a positive feedback and then recruiting RomX and MglA (Fig. 7f, lower panel). Thus, in this configuration, the polarity module is less sensitive to the Frz system, while front-rear polarity is robustly maintained. Based on theoretical arguments, we previously argued that the configuration with a high concentration of the RomR/RomX GEF at the lagging pole would allow for the rapid accumulation of MglA-GTP at this pole in response to Frz signaling, allowing the inversion of polarity. We, therefore, suggest that the spatial configuration of the polarity proteins in the Δ*mglC* mutant makes it less sensitive to Frz signaling because there is too little RomR/RomX GEF at the lagging pole to recruit MglA-GTP during reversals. Thus, the RomR/MglC/MglB positive feedback resulting in the peculiar colocalization of the RomR/RomX GEF and MglB/RomY GAP at the lagging pole in WT serves two purposes: First, the GAP activity displaces MglA-GTP from this pole to enable unidirectional translocation; and, second, the GEF activity is necessary to provide the system with the ability to rapidly and efficiently invert polarity. In other words, an important function of MglC and the RomR/MglC/MglB positive feedback is to establish the configuration of the polarity proteins that confer the polarity module with responsiveness to the Frz system. Returning to the question raised in the introduction, i.e. why different network designs have been selected for in various polarity-regulating networks with functionally equivalent outcomes, in the *S. cerevisiae* polarity system that establishes the single Cdc42 cluster, the positive feedback is centered on Cdc42 and the Cdc24 GEF[9]. Therefore, once the Cdc42 cluster is established, this polarity is stably maintained, and the decay of a nascent bud site or the formation of competing bud sites is efficiently avoided. In the Δ*mglC* mutant, RomR, by stimulating its own polar binding in a positive feedback, brings about RomR/RomX and MglA polar localization at the same pole (Fig. 7f, lower panel). This design is conceptually similar to the yeast system driving Cdc42 cluster formation. Thus, while the network designs of the *M. xanthus* and the *S. cerevisiae* polarity systems enable the formation of a single MglA/Cdc42 cluster, the different wirings can be rationalized as the *M. xanthus* polarity module being part of a spatial toggle switch that is optimal for stable polarity as well for rapid polarity inversions. By contrast, the *S. cerevisiae* system is optimized to provide stable polarity.

In principle, it would seem that the RomR/MglC/MglB positive feedback could have been established by RomR interacting directly

with MglB, raising the question of the advantage of incorporating MglC into the RomR/MglC/MglB positive feedback. The roadblock domain protein family is ancient, abundantly present in all domains of life, and often involved in regulating GTPase activity[37,46–48]. Interestingly, the Rag GTPases of the mTOR pathway are composed of a small GTPase domain and a C-terminal roadblock domain and form heterodimers using their roadblock domains[49]. These heterodimers are recruited to lysosomes by the Ragulator complex, which contains two roadblock heterodimers that interact head-to-tail forming a tetrameric complex[49]. The Rag GTPase/Ragulator interaction occurs via the roadblock domains, resulting in three layers of heterodimeric road-block domains[49]. Intriguingly, Rag heterodimers' GTP/GDP state allosterically regulates their binding to Ragulator by tuning the inter-action between pairs of roadblock heterodimers[50,51]. This mechanism is conceptually remarkably similar to the GTP/GDP state of MglA reg-ulating the interaction between the MglC/MglB homodimers, sup-porting that this regulatory mechanism is evolutionary conserved. We suggest that the presence of MglC in the RomR/MglC/MglB positive feedback reflects an ancient regulatory mechanism in which the GTP/GDP state of a partner GTPase can modulate the interaction between pairs of roadblock dimers.

## Methods

### Cell growth and construction of strains
Strains, plasmids and primers used in this work are listed in Supple-mentary Table 1, 2 and 3, respectively. All *M. xanthus* strains are deri-vatives of the DK1622 WT strain[52]. *M. xanthus* was grown at 32 °C in 1% CTT broth[53] or on 1.5% agar supplemented with 1% CTT and kanamycin (50 µg mL$^{-1}$) or oxytetracycline (10 µg mL$^{-1}$) as appropriate. In-frame deletions were generated as described[54]. Plasmids were introduced in *M. xanthus* by electroporation and integrated by homologous recombination at the endogenous locus or at the *mxan18-19* locus or by site-specific recombination at the Mx8 *attB* site. All in-frame dele-tions and plasmid integrations were verified by PCR. Plasmids were propagated in *Escherichia coli* TOP10 (F⁻, *mcrA*, Δ(*mrr-hsd*RMS-*mcr*BC), φ80*lacZ*ΔM15, Δ*lac*X74, *deo*R, *rec*A1, *ara*D139, Δ(*ara-leu*)7679, *gal*U, *gal*K, *rps*L, *end*A1, *nup*G). *E. coli* was grown in LB or on plates containing LB supplemented with 1.5% agar at 37 °C with added antibiotics when appropriate[55]. All DNA fragments generated by PCR were verified by sequencing.

### Motility assays and determination of reversal frequency
Population-based motility assays were done as described[56]. Briefly, *M. xanthus* cells from exponentially growing cultures were harvested at 4000× *g* for 10 min at room temperature (RT) and resuspended in 1% CTT to a calculated density of 7×10⁹ cells mL$^{-1}$. 5 µL aliquots of cell suspensions were placed on 0.5% agar plates supplemented with 0.5% CTT for T4P-dependent motility and 1.5% agar plates supplemented with 0.5% CTT for gliding motility and incubated at 32 °C. After 24 h, colony edges were visualized using a Leica M205FA stereomicroscope and imaged using a Hamamatsu ORCA-flash V2 Digital CMOS camera (Hamamatsu Photonics) using the LASX software (Leica Micro-systems). For higher magnifications of cells at colony edges on 1.5% agar, cells were visualized using a Leica DMi8 inverted microscope and imaged with a Leica DFC9000 GT camera.

Individual cells were tracked as described[29]. Briefly, for T4P-dependent motility, 5 µL of exponentially growing cultures were spotted into a 24-well polystyrene plate (Falcon). After 10 min at RT, cells were covered with 500 µL 1% methylcellulose in MMC buffer (10 mM MOPS (3-(*N*-morpholino) propanesulfonic acid) pH 7.6, 4 mM MgSO₄, 2 mM CaCl₂), and incubated at RT for 30 min. Subsequently, cells were visualized for 10 min at 20 sec intervals at RT using a Leica DMi8 inverted microscope and a Leica DFC9000 GT camera and the LASX software (Leica Microsystems).. Individual cells were tracked using Metamorph 7.5 (Molecular Devices) and ImageJ 1.52b[57] and then

the speed of individual cells per 20 sec interval as well as the number of reversals per cell per 10 min calculated. For gliding, 5 µL of exponen-tially growing cultures were placed on 1.5% agar plates supplemented with 0.5% CTT, covered by a cover slide and incubated at 32 °C. After 4 to 6 h, cells were observed for 15 min at 30 sec intervals at RT as described, speed per 30 sec interval as well as the number of reversals per 15 min calculated.

### Immunoblot analysis
Immunoblot analysis was done as described[55]. Rabbit polyclonal α-MglA[27] (dilution 1:5000), α-MglB[27] (dilution 1:5000), α-RomR[32] (1:5000), α-PilC[58] (dilution 1:5000), α-PilO[59] (dilution 1:2000), α-PilT[60] (dilution 1:2000), α-mCherry (Biovision, dilution 1:15000) and α-MglC antibodies (dilution 1:5000) were used together with horseradish peroxidase-conjugated goat anti-rabbit immunoglobulin G (Sigma) as a secondary antibody (dilution 1:10000). Mouse α-GFP (Sigma, dilution 1:2000) and α-EF-Tu (HycultBiotech, dilution 1:5000) antibodies were used together with horseradish peroxidase conjugated sheep anti-mouse immunoglobulin G (GE Healthcare) as a secondary antibody (dilution 1:2000). To generate rabbit polyclonal α-MglC antibodies, His₆-MglC was purified as described (see below) and used for immu-nization as described[55]. Blots were developed using Luminata Cres-cendo Western HRP substrate (Millipore) and visualized using a LAS-4000 luminescent image analyzer (Fujifilm). Proteins were separated by SDS-PAGE as described[55].

### Fluorescence microscopy
For fluorescence microscopy, exponentially growing cells were placed on slides containing a thin pad of 1% SeaKem LE agarose (Cambrex) with TPM buffer (10 mM Tris-HCl pH 7.6, 1 mM KH₂PO₄ pH 7.6, 8 mM MgSO₄) and 0.2% CTT, and covered with a coverslip. After 30 min at 32 °C, cells were visualized using a temperature-controlled Leica DMi8 inverted microscope and phase contrast and fluorescence images acquired using a Hamamatsu ORCA-flash V2 Digital CMOS camera and the LASX software (Leica Microsystems). For time-lapse recordings, cells were imaged for 15 min using the same conditions. To induce expression of genes from the vanillate inducible promoter[61], cells were treated as described in the presence of 300 µM vanillate and imaged for 6 h. To precisely quantify the localization of fluorescently labelled proteins, we used an established analysis pipeline[17] in which the output for each cell is total cellular fluorescence, total fluorescence in clusters at each pole and the mean fractions of fluorescence at each of the two poles. Data points for individual cells were plotted in scatterplots in which the poles with the highest and lowest polar fraction of fluores-cence are defined as pole 1 and pole 2, respectively. Dispersion of the single-cell measurements is calculated and shown by error bars and ellipses in which the direction and length of error bars are defined by the eigenvectors and square root of the corresponding eigenvalues of the polar fraction covariance matrix for each strain. For calculating mean fraction of fluorescence at the poles, cells with and without clusters were included. The quantification of fluorescence signals is included in Supplementary Table 4.

### Image analysis
Microscope images were processed with Fiji[62] and cell masks determined using Oufti[63] and manually corrected when necessary. Fluorescence was quantified in Matlab R2020a (The MathWorks) as described[17].

### In vivo fluorescence recovery after photobleaching (FRAP)
FRAP experiments were performed as described[64] with a temperature-controlled Nikon Ti-E microscope with Perfect Focus System and a CFI PL APO 100x/1.45 Lambda oil objective at 32 °C with a Hamamatsu Orca Flash 4.0 camera using NIS Elements AR 2.30 software (Nikon) in the dark. Photobleaching was performed using a single circular shaped

region with 20% laser power (561 nm) and a 500 μsec dwelling time. For every image, integrated fluorescence intensities of a whole cell and the bleached region of interest (ROI) were measured. After background correction, the corrected fluorescence intensity of the bleached ROI was divided by total corrected cellular fluorescence, correcting for bleaching effects during picture acquisition. Cell segmentation and background correction was performed with Oufti. This normalized fluorescence was correlated to the initial fluorescence in the ROI. The mean relative fluorescence of cells was plotted as a function of time. The recovery rate for a given fluorescent protein was determined by fitting the plotted data to a single exponential equation with Matlab R2020a (The MathWorks).

## Protein purification

All proteins were expressed in *E. coli* Rosetta 2(DE3) (F$^-$ *ompT hsd*S$_B$ (r$_B$$^-$ m$_B$$^-$) *gal dcm* (DE3 pRARE2) at 18 °C or 37 °C. To purify His$_6$-tagged proteins, Ni-NTA affinity purification was used. Briefly, cells were washed in buffer A (50 mM Tris-HCl pH 7.5, 150 mM NaCl, 10 mM imidazole, 5% glycerol, 5 mM MgCl$_2$) and resuspended in lysis buffer A (50 mL of wash buffer A supplemented with 1 mM DTT, 100 μg ml$^{-1}$ phenylmethylsulfonylfluoride (PMSF), 10U ml$^{-1}$ DNase 1 and Complete Protease Inhibitor Cocktail Tablet (Roche)). Cells were lysed by sonication and cell debris was removed by centrifugation (48,000× *g*, 4 °C, 30 min) and filtration through a 0.45 μm filter (Sarsted). The cleared cell lysate was loaded onto a 5 mL HiTrap Chelating HP column (Cytiva) preloaded with NiSO$_4$ as described by the manufacturer and pre-equilibrated in buffer A. The column was washed with 20 column volumes of column wash buffer (buffer A with 20 mM imidazole). Proteins were eluted with elution buffer (buffer A with 500 mM imidazole) using a linear imidazole gradient from 20 to 500 mM. Fractions containing purified His$_6$-tagged proteins were combined and loaded onto a HiLoad 16/600 Superdex 75 pg (GE Healthcare) gel filtration column that was equilibrated with buffer 1 (50 mM Tris-HCl pH 7.5, 150 mM NaCl, 1 mM DTT, 5 mM MgCl$_2$, 5% glycerol). Fractions containing His$_6$-tagged proteins were pooled, frozen in liquid nitrogen and stored at −80 °C.

To purify MalE-tagged proteins (MalE-RomR, MalR-RomR$^{1-368}$, MalR-RomR$^C$ and MalE-MglC), maltose-binding protein (MBP) affinity purification was used. Briefly, cells were washed in buffer B (50 mM Tris-HCl pH 7.5, 150 mM NaCl, 1 mM EDTA, 1 mM DTT) and resuspended in 50 mL lysis buffer B (50 mL buffer B supplemented with PMSF 100 μg mL$^{-1}$, DNase 1 10U mL$^{-1}$ and Complete Protease Inhibitor Cocktail Tablet (Roche)). Cells were lysed and cleared cell lysates prepared as described and loaded onto a 5 mL MBPTrapHP (Cytiva) column equilibrated with buffer B. The column was washed with 20 column volumes of buffer B. Proteins were eluted with elution buffer B (buffer B with 10 mM maltose). Eluted fractions containing the relevant MalE-tagged protein were loaded onto a 5 mL HiTrap Q HP ion exchange column (Cytiva) equilibrated with buffer C (50 mM Tris-HCl pH 7.5, 50 mM NaCl, 5 mM MgCl$_2$, 1 mM DTT, 5% glycerol). The column was washed with 20 column volumes of buffer C. The Mal-tagged proteins were eluted with buffer C using a linear gradient of NaCl from 50 to 500 mM. Fractions containing the MalE-tagged proteins were loaded onto a HiLoad 16/600 Superdex 200 pg (GE Healthcare) gel filtration column that was equilibrated with buffer 1. Fractions with MalE-tagged proteins were pooled, frozen in liquid nitrogen and stored at −80 °C.

To purify Strep-tagged proteins, biotin affinity purification was used. Briefly, cells were washed in buffer C (100 mM Tris-HCl pH 8.0, 150 mM NaCl, 1 mM EDTA, 1 mM DTT) and resuspended in lysis buffer C (50 mL of wash buffer C supplemented with 100 μg mL$^{-1}$ PMSF, 10U mL$^{-1}$ DNase 1 and Complete Protease Inhibitor Cocktail Tablet (Roche)). Cells were lysed and cleared lysate prepared as described and loaded onto a 5 mL Strep Trap HP (Cytiva) column, equilibrated with buffer C. The column was washed with 20 column volumes of buffer C. Protein was eluted with elution buffer C (buffer C with 2.5 mM

desthiobiotin). Elution fractions containing Strep-tagged proteins were loaded onto a HiLoad 16/600 Superdex 75 pg (GE Healthcare) gel filtration column that was equilibrated with buffer 1. Fractions with Strep-tagged proteins were pooled, frozen in liquid nitrogen and stored at −80 °C.

## Pull-down experiments

To test for interactions with MglA, MglB and RomR variants, Strep-MglC (final concentration 10 μM) was incubated with MglA-His$_6$, His$_6$-MglB, MalE-RomR, MalE-RomR$^{1-368}$ or MalE-RomR$^C$ (final concentration 10 μM) in buffer 1 (50 mM Tris-HCl pH 7.5, 150 mM NaCl, 1 mM DTT, 5 mM MgCl$_2$, 5% glycerol) for 30 min at RT. Subsequently, 10 μL of Strep-Tactin MagStrep' type3' XT beads (IBA Lifesciences) previously equilibrated with buffer 1 was added for 30 min at RT. Then beads were washed 10 times with 1 mL buffer 1. Proteins were eluted with 200 μL elution buffer (100 mM Tris-HCl pH 8.0, 150 mM NaCl, 1 mM EDTA, 50 mM biotin). To test for interactions with MglC, MglA and MglB, MalE-RomR (final concentration 10 μM) was incubated with Strep-MglC, MglA-His$_6$ and/or His$_6$-MglB (final concentration 10 μM) in buffer 1 for 30 min at RT. Subsequently, the mixture was added to 200 μL of Amylose Resin, previously equilibrated with buffer 1, and incubated for 30 min at RT. The resin was then washed 10 times with 1 mL buffer 1. Proteins were eluted with 200 μL elution buffer (100 mM Tris-HCl pH 8.0, 150 mM NaCl, 1 mM EDTA, 10 mM amylose). To test for interactions with MglC, MglA and RomR, His$_6$-MglB (final concentration 10 μM) was incubated with Strep-MglC, MglA-His$_6$ and/or MalE-RomR (final concentration 10 μM) in buffer 1 for 30 min at RT. Subsequently, 20 μL of Amintra Nickel Magnetic beads (Expedeon), previously equilibrated with buffer 1, was added to the mixture and incubated for 30 min at RT. Beads were then washed 10 times with 1 mL buffer 2 (buffer 1 with 50 mM imidazole). Proteins were eluted with 200 μL elution buffer (buffer 1 with 500 mM imidazole). In experiments involving MglA-His$_6$, MglA-His$_6$ (final concentration 10 μM) was preloaded with GTP, GDP or GppNHp (final concentration 40 μM) for 30 min at RT in buffer 1 and all buffers contained 40 μM of the relevant nucleotide.

## GTPase assays

GTP-hydrolysis by MglA-His$_6$ was measured using a continuous, regenerative coupled GTPase assay[65] in reaction buffer (50 mM Tris-HCl pH 7.5, 150 mM NaCl, 5% glycerol, 1 mM DTT, 7.5 mM MgCl$_2$) supplemented with 495 μM NADH (Sigma), 2 mM phosphoenolpyruvate (Sigma), 18-30U mL$^{-1}$ pyruvate kinase (Sigma) and 27-42 U mL$^{-1}$ lactate dehydrogenase (Sigma). For all assays, MglA-His$_6$ (final concentration 2 μM) was preloaded with GTP (final concentration 3.3 mM) for 30 min at RT in reaction buffer. In parallel, MglB was preincubated with Strep-MglC, MalE-RomR and/or Strep-RomY for 10 min at RT in reaction buffer. GTPase reactions were performed in 96-well plates (Greiner Bio-One) and initiated by adding His$_6$-MglB, Strep-MglC, MalE-RomR and/or Strep-RomY to the MglA/GTP mixture. Final concentration, MglA-His$_6$: 2 μM, His$_6$-MglB: 4 μM, Strep-MglC: 4 μM, MalE-RomR: 2 μM, Strep-RomY: 2 μM, GTP: 1 mM. Absorption was measured at 340 nm for 60 min at 37 °C using an Infinite M200 Pro plate-reader (Tecan) and the amount of hydrolyzed GTP per hour per molecule of MglA-His$_6$ calculated. For each reaction, background subtracted GTPase activity was calculated as the mean of three technical replicates.

## Solubility test

To test the solubility of MglC-mVenus, MglC$^{FDI}$-mVenus and MglC$^{KRR}$-mVenus in *M. xanthus* cells were fractionated into fractions enriched for soluble and insoluble proteins, including inner membrane and outer membrane proteins as described[66]. Briefly, exponentially growing cultures were harvested at 11,000 *g* for 10 min at RT and the cell pellet resuspended in Lysis Buffer D (50 mM Tris-HCl pH 7.5, 150 mM NaCl, 5 mM MgCl$_2$, supplemented with Complete Protease Inhibitor

Cocktail Tablet (Roche)). Cells were lysed using sonication and lysates cleared by centrifugation at 8000 *g* for 5 min, RT. The cleared lysate was subjected to ultra-centrifugation using an Air-Fuge (Beckman) at ~150,000 *g* for 1 hr. The resulting pellet contains insoluble proteins and was separated from the supernatant, which contains soluble proteins, and resuspended in Lysis Buffer D. Both fractions were mixed with SDS-lysis buffer and analyzed by by SDS-PAGE and immunoblotting. As a control for a protein that forms inclusion bodies, $His_6$-PilT[60] expressed in *E. coli* was used. EF-Tu was used as a control for a soluble *E. coli* protein. *E. coli* cells were treated as *M. xanthus* cells.

## Mass photometry (MP)

MP was performed using a TwoMP mass photometer (Refeyn Ltd, Oxford, UK). Data acquisition was performed using AcquireMP (Refeyn Ltd. v2.3). MP movies were recorded at 1 kHz, with exposure times varying between 0.6 and 0.9 ms, adjusted to maximize camera counts while avoiding saturation. Microscope slides (1.5 H, 24×50 mm, Carl Roth) and CultureWellTM Reusable Gaskets were cleaned with three consecutive rinsing steps of double-distilled $H_2O$ and 100% iso-propanol and dried under a stream of pressurized air. For measurements, gaskets were assembled on coverslips and placed on the stage of the mass photometer with immersion oil. Assembled coverslips were held in place using magnets. For measurements, gasket wells were filled with 10 μL of 1× phosphate-buffered saline (137 mM NaCl, 2.7 mM KCl, 8 mM $Na_2HPO_4$, 2 mM $KH_2PO_4$) to enable focusing of the glass surface. After focusing, 10 μL sample were added, rapidly mixed while keeping the focus position stable and measurements started. MP contrast values were calibrated to molecular masses using an in-house standard. For each sample, three separate measurements were performed. The data were analyzed using the DiscoverMP software (Refeyn Ltd, v. 2022 R1). MP image analysis was done as described[67].

## AlphaFold model generation

AlphaFold-multimer (version 2.3.1) structure prediction was done with the ColabFold pipeline[43-45]. ColabFold was executed with default settings where multiple sequence alignments were generated with MMseqs2[68] and HHsearch[69]. The ColabFold pipeline generates five model ranks. Predicted Local Distance Difference Test (pLDDT) and alignment error (pAE) graphs were generated for each rank with a custom Matlab R2020a (The MathWorks) script. Ranking of the models was performed based on combined pLDDT and pAE values, with the best-ranked models used for further analysis and presentation. Per residue model accuracy was estimated based on pLDDT values (>90, high accuracy; 70-90, generally good accuracy; 50-70, low accuracy; <50, should not be interpreted)[45]. Relative domain positions were validated by pAE[45]. Only models of the highest confidence, based on combined pLDDT and pAE values, were used for further investigation. For all models, sequences of full-length proteins were used.

## Bioinformatics

Sequence alignments were generated using ClustalOmega[70] with default parameters and alignments were visualized with Jalview[71]. Protein domains were identified using Interpro[72]. Charge score was calculated using the Protein-sol tool[73]. Structural alignments and calculation of electrostatic surface potential were done in Pymol (The PyMOL Molecular Graphics System, Version 1.2r3pre, Schrödinger, LLC).

## Statistics

Statistics were performed using a two-tailed Student's *t*-test for samples with unequal variances.

## Reporting summary

Further information on research design is available in the Nature Portfolio Reporting Summary linked to this article.

## Data availability

The structural data mentioned in this study is available in the Protein Data Bank database under the accession codes 6HJM (MglB), 7CY1 (MglC) and 6IZW (MglA bound to GTP-γ-S and MglB). The authors declare that all data supporting this study are available within the article, its Supplementary Information file or in the source data file. Source data are provided with this paper.

## Code availability

Custom MATLAB scripts used for analysis of fluorescence data and FRAP data are available from the corresponding author upon request.

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

## Acknowledgements

We thank Dr. Anna McLoon for the generation of the α-MglC antibodies. G.K.A.H. and L.S.-A. gratefully acknowledge the financial support by the Max Planck Society.

## Author contributions

Conceptualization: L.A.M.C. and L.S.-A. Experimental work: L.A.M.C., D.S. and S.L. Analysis of experimental data: L.A.M.C. and S.L. Writing – original draft: L.A.M.C. and L.S.-A. Writing – editing of draft: L.A.M.C., D.S., S.L., G.K.A.H. and L.S.-A. Supervision: G.K.A.H. and L.S.-A. Funding acquisition: G.K.A.H. and L.S.-A.

## Funding

## Competing interests

The authors declare no competing interests.
