## [Peer Review File · Nature Communications]

REVIEWER COMMENTS

Reviewer #1 (Remarks to the Author):

This manuscript clarified the function of MglC and elucidated a positive feed back by the RomR/MglC/MglB complex that establishes the lagging pole in *M. xanthus*. Overall, the experiments were well designed and executed. The authors did a great job in explaining such a complicated regulatory mechanism.

I have the following comments regarding the manuscript:

1. The authors were not able to purify soluble MglC FDI and KRR from *E. coli*. However, they showed the localization patterns of both variants in live *M. xanthus* cells. A control experiment is required to show that these variants, especially KRR, whose localization pattern totally collapses, are soluble in *M. xanthus*.

2. It is very intriguing to see that RomR binds to the MglB/C complex as both dimers and trimers. While the authors showed good evidence that RomR predominantly exists as trimers in vivo, the same evidence does not support the authors' claim that "trimeric RomR is the active form" (L434-435). Similarly, the authors showed nice, negative evidence that the C-terminus truncated form of RomR (RomR1-368) is unable to form trimers, localize to poles, or interact with MglC. However, these data do not necessarily support the roles of RomR-C in oligomerization, polar localization, and interaction with MglC. Rather, losing the C-terminal domain could have totally collapsed the structure of RomR, making the truncated protein nonfunctional. To support the authors' claim, positive evidence should be shown, in which the C-terminal domain of RomR alone is able to form trimers, localize to poles, and interact with MglC.

3. L83-87 in the introduction needs to be clarified. Why "MglB alone has GAP activity" but "the MglB/RomR complex is the active GAP"? Why "RomX alone has GEF activity" but "the RomR/RomX complex is the active GEF"?

Reviewer #2 (Remarks to the Author):

The manuscript by Carriera et al attempts to provide a mechanism for the front-rear polarity switch based on the establishment of the role of MglC in driving the cooperative accumulation of the lagging pole components mainly RomR and MglB. The authors propose that the formation of this complex and the positive feedback loop results in high GAP activity at the leading pole thus eliminating MglA accumulation at the lagging pole. Authors have included genetics experiments for providing localization information, biochemical evidence including a thorough combination of pull down experiments, proposed structural model using AlphaFold and corroborative mutational data to support their hypothesis based on the AlphaFold prediction.

I have concerns about the derivation of the model, both with respect to the mutations predicted to be at the interface, the hypothesis based on the AlphaFold model and the proposed allostery based mechanism of the how MglA is driven away from the complex. Concerns about the technical details and the data interpretations are detailed below.

Major points:

1. Mutational analysis

a) MglB-KRK mutant:

The KRK surface does not face the proposed model of the MglC interface. MglBKRK is not within an interaction distance with the residues of MglC according to the predicted structure of the complex. How was it hypothesized that these could play a role? What is the explanation for how these mutations could result in a complete disruption of the complex?

Since this MglB mutant does not affect MglA GAP activity (Galicia et al, 2019), why should it affect MglA-mVenus localization (line 263)? If the MglA localization is affected, it should imply that the interaction between MglA and MglB is affected?

b) MglC-FDI mutant

Line 275-276: Given the difficulties faced in purification of MglCFDI complex, how have the authors ensured that the corresponding mutation results in a folded and functional protein within the Myxococcus cells? Could it be that the effect observed was due to the partial unfolding of MglC resulting in a proportion of non-functional protein?

All the observed phenotypes of the mutant (lines 275 - 283) are equivalent to those of a deletion of MglC? Also in Supp Fig. 5, the amount of MglC-FDI-mVenus is quite weak compared to that of MglC-WT-mVenus.

c) MglC-KRK mutant:

Same question as above is relevant for MglC-KRR as well. Are these signatures of unfolded non-functional protein rather than specific effects of the mutation?

Though I do understand that it is difficult to confirm the folded state of the mutants in vivo, the authors have to be cautious in interpreting the role of these residues. This can only be a speculation, but not a strong proof. Can an MglB-R115 or MglC-D mutant instead of the triple mutants be purified and tested in vitro and in vivo? This might provide the experimental proof for the hypotheses proposed by the authors.

2. RomR oligomerization data:

Lines 304 - 306: Will the complementary construct (369 - 420 of RomR) form a stable trimer? Have the authors checked this possibility?

The impurities in the RomR purification makes the MP data inconclusive and the trimer formation shaky. Use of SEC-MALS coupled with checking the eluted fractions on SDS-PAGE will be a more reliable method for confirming the trimer formation (and complex formation as well - refer next point) in the absence of a pure preparation of MalE-RomR protein sample.

The quantitative immunoblot analysis data: please include the reference to a citation or an appropriate figure. I could not find where the data has been presented.

Line 434 - 435 - Couldn't find the data for this observation. Please clarify and cite the relevant figure number. Correct Rom-C to RomR-C

3. RomR-MglC interaction:

The higher molecular weights in MP can be interpreted in multiple possibilities (for example, 297 could be RomR trimer plus MglC monomer given the surprisingly high level of monomeric MglC, which is expected to be an obligate dimer similar to MglB). In the absence of assays similar to gel filtration, which will demonstrate a shift as well as the eluted fractions can be confirmed on an SDS-PAGE gel to establish presence of the complex, the data from MP is not convincing.

Again, can this be established using a RomR 369 - 420 construct?

Also, the stoichiometry is not very compatible with the symmetry of MglC, where the proposed interacting residues are at the symmetric dimeric interface, and it is difficult to imagine how a trimer of RomR could fit in here.

4. Allosteric model based on AlphaFold prediction:

Line 424-426: This allosteric mechanism is questionable. AlphaFold does not provide a proof for it, but only a weak speculation. The dimeric MglB structure generated from AlphaFold also matches with that of the MglB/MglC structure with similar helix orientations, whereas the experimental MglB dimer structure (6HJM) resembles that of MglAB complex (6IZW). Hence, it is possible that the differences in orientation is an artefact of AlphaFold prediction, and may or may not reflect an allostery based mechanism.

Superposition of MglB dimer structure with 6IZW is an overall superposition of the dimer. The best demonstration of a relative movement between the two monomers of MglB comes up when one of the protomers is superposed. Such a superposition shows that there is an angular difference between the two protomers in the AlphaFold generated MglB dimer, compared to the structures of MglB with MglA (6IZW) or MglB alone (6HJM). Redo the figure panel in the Supplementary accordingly.

5. Positive feedback loop:

The positive feedback loop assumes that RomR recruits C and then B. However,

a mechanism based on breaking the interface between MglC and MglB implies that the levels of MglB will not be commensurate with that of RomR, both of which are localised at the lagging pole. Won't such a model lead to a loss of MglB from the lagging pole too, and not only detachment of MglA? How can a RomR/MglC/MglB positive feedback function in this case?

Can the FRAP experiments be extended to labeled MglC and MglB also to provide proof for the positive feedback?

In panel 7A (2nd gel), can the observation be explained by the fact that MglB has been sequestered by MglA? Has the complex formation of MglAB been monitored by an equivalent experiment with a tag on MglA or MglB?

Similarly, what is the explanation of minor binding in the 3rd panel? Here too, MglA is capable of binding to MglB with GDP as well (Baranwal, et al, 2019).

6. GAP and GEF activities at the lagging pole:

Lines 467 - 469: Is it possible that the RomRX GEF activity is not active or relevant at the lagging pole, but is required only at its low levels in the leading pole? The idea of both GAP and GEF being necessary at the lagging pole is not clear. The system should be an efficient GAP and an inefficient GEF at the lagging pole for MglA localization to be correctly established?

Supp Fig 4: According to the allosteric mechanism, the presence of all the components should either break the interface between MglA and MglB and hence reduce the GAP activity? How can the authors claim high GAP activity to drive the MglA from the lagging pole, given that the allosteric model is valid?

Line 440: It is proposed in Szadkowski et al, 2022, that RomyY interacts with MglB and an MglA-GTP state. Since MglA-GTP-MglB and MglB-MglC are mutually exclusive, is it correct propose a RomR/MglC/MglB.RomY complex?

Minor points:

1. Lines 174 - 178: How is it concluded that the localization strongly depends on MglC? This conclusion can be made only in a background where only MglC is deleted? In delta MglC, MglB-mCherry was strongly reduced (line 151).
2. Line 181-182: The basis of this conclusion that 'neither MglC or MglB alone stimulates polar localization of RomR' is not convincing given the decrease in RomR upon deletion of MglB or MglC.
3. Line 184: How is the order of recruitment derived based on the observations? It need not be the only option.
4. The final R for RomR is missing in a couple of instances in the Discussion.

Reviewer #3 (Remarks to the Author):

Regulated reversals of the direction of individual cell movement enable populations of *Myxococcus xanthus* bacteria to coordinate multicellular behaviors such as predatory feeding and starvation-induced fruiting body development. Elucidating the regulation of cell reversal is key to understanding the multicellular behaviors of *M. xanthus* and, more broadly, the mechanisms by which cells dynamically control polar localization of proteins. Many proteins involved in *M. xanthus* polarity control have been described, but important mechanistic and design questions have remained. The authors report that MglC interacts with RomR and MglB at the rear pole of the cell, where GAP activity of MglB/RomY inhibits MglA-GTP accumulation and assembly of motility machineries. They also provide evidence that MglA-GTP at the front pole inhibits the interaction of MglB with MglC. Although RomR/RomX GEF activity at the rear pole would tend to stimulate MglA-GTP accumulation, thus counteracting

MglB/RomY inhibition, this design appears to poise the system for rapid reversal upon signaling from the Frz chemosensory system. The authors compare this design with two others involving GTPase activity; yeast polar budding with a simpler design, and mammalian cell growth regulation with comparable complexity to *M. xanthus* in terms of GTP/GDP control of protein-protein interactions.

The work significantly advances understanding of the cell polarity regulatory network in *M. xanthus*. Previously, the Sogaard-Andersen group identified MglC as a paralog of MglB and used phenotypic and genetic approaches to begin characterizing MglC function (McLoon et al. 2016). Here, the authors use a battery of approaches to take the work much further by quantifying polar localization of functional fluorescent fusions to MglC, MglB, RomR, and MglA in a large number of single, double, and triple knockout mutants, purifying tagged proteins and performing biochemical pull-down assays (including with protein variants to identify interacting surfaces), characterizing oligomerization and functions of the RomR C-terminal domain, using FRAP to show that MglC and MglB stabilize RomR polar localization, using pull-downs to show that MglA-GTP breaks the MglC/MglB interaction but not the MglC/RomR interaction, and proposing a model of the regulatory interactions at the front and rear poles.

The work supports the conclusions and in general is very well done. There is room for some improvements (see comments below), but this should not prohibit publication. The methodology is sound and well-described, and in general the manuscript is very well-written for a broad audience.

Comments:

The Abstract (l. 31-34) is difficult to understand because the positive and negative feedback are not adequately explained. The authors state “that RomR and the MglB and MglC roadblock domain proteins generate a positive feedback by forming a RomR/MglC/MglB complex”, but this fails to explain the nature of the positive feedback. It seems better to say “that RomR and the MglB and MglC roadblock domain proteins enhance each other’s polar localization by forming a RomR/MglC/MglB complex, ...”. Likewise, the following sentence would be easier to understand as “MglA at the front inhibits formation of the RomR/MglC/MglB complex by binding to MglB and breaking its interaction with MglC, ...”. The positive and negative feedback could then be explained in the Introduction if the authors want to use those terms.

l. 84 – the authors state that “MglB alone has GAP activity” and they never mention that it has GEF activity, yet ref. 41 states that “The dual GAP-GEF activities of MglB accelerate the rate of GTP hydrolysis over multiple enzymatic cycles.” Why don’t the authors mention the GEF activity of MglB?

I. 129 – a gliding motility defect of the *mgIC* mutant is not evident from Fig. S1A since there appear to be as many single cells as for WT at the edge of the colony on 1.5% agar. However, I'm not sure what are single cells (arrows may help) even when viewed at 800%.

Fig. S1C – the *frzE* mutant appears to move faster than WT on 1.5% agar. Was this known? Why would it be the case?

Fig. S1C - why is the speed and reversal not measured for the complemented *mgIC* mutant and the *MgIC-mVenus* strain?

Fig. 1C – in the panel showing *MgIA-mVenus*, the slope of the gray lines is not -1, as in the other panels. Is there a mistake?

I. 275-278 – this part is weak. The authors state they were unable to purify a soluble Strep-tagged *MgIC* FDI variant. Did the protein accumulate? Did the authors try to overcome this problem by changing only D to A, or only D to K (charge reversal), since F and I might contribute to the hydrophobic core such that A at both those positions leads to unfolding, and only D contributes to the negatively charged surface? The other weakness is that *MgIC* FDI-*mVenus* accumulates poorly relative to *MgIC-mVenus* (Fig. S5B), which could affect the polar localization of the variant. Did the authors try to increase *MgIC* FDI-*mVenus* accumulation by fusing the gene to the *pilA* promoter, which appeared to increase the *MgIC* FDI level in the *mgIC* mutant relative to the *MgIC* level in WT (Fig. S5B)? If these issues cannot be resolved or the authors do not wish to do additional experiments, Fig. 4 could be moved to the Supplementary Information.

I. 291-292 – as in my preceding comment, this part is weakened by inability to purify the *MgIC* variant, which is surprising given that KRR are presumably surface-exposed residues. Similar questions as in the preceding comment arise. Did the protein accumulate? Did the authors try to overcome this problem by changing only one or two of the three charged residues? This part is not as weak as in the preceding comment, since in this case the *MgIC* KRR-*mVenus* level was similar to that of *MgIC-mVenus* (Fig. S5C), so the level of the variant should not affect polar localization. However, the *romR* single mutant was not tested for *MgIC* KRR-*mVenus* localization (Fig. 4D). Granted, localization would very likely be lost since that was the case for *MgIC-mVenus* (Fig. 1E), but it seems odd to leave it out since the authors infer that KRR of *MgIC* interacts with *RomR*. In any case, the main difference between the FDI and KRR variants of *MgIC* is the effects of the *mgIB* and *mgIA* single mutants in panels B and D of Fig. 4, and I'm not convinced that this observation warrants two sections of Results and Fig. 4. The evidence for the inferred interaction surfaces of *MgIC* with *MgIB* and *RomR* is weak and could be moved to the Supplementary Information without detracting from the main conclusions of the manuscript.

Fig. 5C – the authors should offer an explanation for the multiple species below MalE-RomR 1-368, perhaps in the legend. Presumably these are proteolytic fragments of the truncated fusion protein.

Fig. 7F top part – does not depict RomR associated with the cell envelope at the pole, yet RomR is crucial for polar localization of the other proteins and the authors state that polar RomR recruits MglC (l. 414). If the arrow from RomR to itself, touching the cell envelope, is meant to depict positive autoregulatory cell envelope association, this should be explained in the legend and the corresponding arrow at the leading pole should touch the cell envelope.

At the end of the Discussion, the authors very nicely come back to the yeast polar budding mentioned in the Introduction and compare it with their findings about *M. xanthus* polarity control. They also make an interesting comparison with mammalian cell growth regulation, which is comparable in complexity to *M. xanthus* motility regulation and likewise involves GTP/GDP control of protein-protein interactions. Perhaps due to a word limit, the authors did not compare their findings with other well-studied examples of regulation of bacterial cell polarity (e.g., *Caulobacter* swarmer versus stalk cell differentiation, *M. xanthus* cell division, etc.). The authors are expert in this area (e.g., Treuner-Lange and Sogaard-Andersen 2014 *J Cell Biol*, Schumacher and Sogaard-Andersen 2017 *Ann Rev Microbiol*), so further comparison of design principles (mentioned in the Title) would broaden interest and likely be preferable to the two sections of Results and Fig. 4 mentioned in my preceding comments, which could be moved to the Supplementary Information.

Minor points:

l. 193 – add “(Fig. 1F)” after MglC

Fig. 1H – typo in the y-axis label – should be “highest”

Fig. S3 legend mentions black circles but they are missing

Fig. 4C legend – “reduces” should be “reduced”

l. 285 – “interphase” should be “interface”

Fig. 5B legend – “fittet” should be “fitted”

Fig. 7C legend – very confusing because not all the colors are explained and “(red/brown)” seems to describe the MglC2:MglB2 model in reverse order. The colors in the inset also need to be explained better.

l. 413 – “KRR” should be “KRK”

l. 475 – “centred” should be “centered”

REVIEWER COMMENTS

Reviewer #1 (Remarks to the Author):

This manuscript clarified the function of MglC and elucidated a positive feed back by the RomR/MglC/MglB complex that establishes the lagging pole in *M. xanthus*. Overall, the experiments were well designed and executed. The authors did a great job in explaining such a complicated regulatory mechanism.

Response: Thank you very much for the very positive, helpful & constructive feedback.

I have the following comments regarding the manuscript:

1. The authors were not able to purify soluble MglC FDI and KRR from *E. coli*. However, they showed the localization patterns of both variants in live *M. xanthus* cells. A control experiment is required to show that these variants, especially KRR, whose localization pattern totally collapses, are soluble in *M. xanthus*.

Response: Great point & thanks! We followed the advice of the reviewer and included in the main text that the MglC^{FDI} and MglC^{KRR} form inclusion bodies when overexpressed in *E. coli* (line 286, 311). Moreover, we included new experiments in which we tested the MglC^{FDI}-mVenus and MglC^{KRR}-mVenus for solubility in *M. xanthus*. We find that both proteins are soluble. These new experiments are described in line 287-288 and 312-313 and included in Supplementary Fig. 5b. We would also like to add that both mVenus-tagged proteins are fluorescent in *M. xanthus* demonstrating that the tag is folded. Moreover, MglC^{FDI}-mVenus is polarly localized in a manner that (just like the WT variant) depends on RomR.

2. It is very intriguing to see that RomR binds to the MglB/C complex as both dimers and trimers. While the authors showed good evidence that RomR predominantly exists as trimers in vivo, the same evidence does not support the authors' claim that "trimeric RomR is the active form" (L434-435). Similarly, the authors showed nice, negative evidence that the C-terminus truncated form of RomR (RomR1-368) is unable to form trimers, localize to poles, or interact with MglC. However, these data do not necessarily support the roles of RomR-C in oligomerization, polar localization, and interaction with MglC. Rather, losing the C-terminal domain could have totally collapsed the structure of RomR, making the truncated protein nonfunctional. To support the authors' claim, positive evidence should be shown, in which the C-terminal domain of RomR alone is able to form trimers, localize to poles, and interact with MglC.

Response: Once again: Great point & thanks! To provide positive evidence that RomR^C alone is sufficient to form dimers and trimers, interact with MglC and localize polarly in *M. xanthus*, we included new experiments in which we analyze RomR^C in more details. Specifically, we generated a MalE-RomR^C variant for *in vitro* analyses and find (line 333-334; Fig. 5b) that MalE-RomR^C in mass photometry experiments was detected with masses matching well monomers, dimers and trimers. Moreover, we find in pull-down experiments that MalE-RomR^C interacts with Strep-MglC (line 343-344; Fig. 5c). Finally, as described in line 349-351, we expressed the RomR^C-mCherry variant in *M. xanthus*. Even when this variant was expressed from the strong *pilA* promoter, it accumulated at a strongly reduced level (Supplementary Fig. 5f); nevertheless, most cells had a weak polar signal (Fig. 5f). Because the RomR^C-mCherry variant accumulated at a strongly reduced level, we did not test whether RomR^C is sufficient to recruit MglC and MglB to the poles. Altogether, these new experiments (together with the experiments with the RomR¹⁻³⁶⁸ variants) allow us to conclude (line 352-354) that the negatively charged RomR^C has

three functions: It is required and sufficient for RomR oligomerization, represents the RomR interface to MglC, and is required and sufficient for the polar localization of RomR.

Concerning the comment in line 434-435 (now line 495-497), we modified the text to say that “*In vivo* quantification of the RomR concentration suggests that trimeric RomR is the predominant form *in vivo*; however, which of the two forms represent the active form remains to be determined”.

3. L83-87 in the introduction needs to be clarified. Why "MglB alone has GAP activity" but "the MglB/RomR complex is the active GAP"? Why "RomX alone has GEF activity" but "the RomR/RomX complex is the active GEF"?

Response: We apologize for not being clearer. We changed the text to “The homodimeric roadblock domain protein MglB alone has GAP activity and together with its low-affinity co-GAP RomY, forms the MglB/RomY complex with even higher GAP activity²⁶⁻²⁸. RomX alone has GEF activity and forms the RomR/RomX complex with even higher GEF activity and also serves as a polar recruitment factor for MglA-GTP²⁹” (line 83-87).

Reviewer #2 (Remarks to the Author):

The manuscript by Carrieri et al attempts to provide a mechanism for the front-rear polarity switch based on the establishment of the role of MglC in driving the cooperative accumulation of the lagging pole components mainly RomR and MglB. The authors propose that the formation of this complex and the positive feedback loop results in high GAP activity at the leading pole thus eliminating MglA accumulation at the lagging pole. Authors have included genetics experiments for providing localization information, biochemical evidence including a thorough combination of pull down experiments, proposed structural model using AlphaFold and corroborative mutational data to support their hypothesis based on the AlphaFold prediction.

I have concerns about the derivation of the model, both with respect to the mutations predicted to be at the interface, the hypothesis based on the AlphaFold model and the proposed allostery based mechanism of the how MglA is driven away from the complex. Concerns about the technical details and the data interpretations are detailed below.

Response: Thank you very much for the very positive, helpful & constructive feedback.

Major points:

1. Mutational analysis

a) MglB-KRK mutant:

The KRK surface does not face the proposed model of the MglC interface. MglBKRK is not within an interaction distance with the residues of MglC according to the predicted structure of the complex. How was it hypothesized that these could play a role? What is the explanation for how these mutations could result in a complete disruption of the complex?

Since this MglB mutant does not affect MglA GAP activity (Galicía et al, 2019), why should it affect MglA-mVenus localization (line 263)? If the MglA localization is affected, it should imply that the interaction between MglA and MglB is affected?

Response: We apologize for not being clearer about how we picked the MglB^{KRK} variant. To address this point, we rewrote the description of this variant taking great care to reference the Galicía et al. paper (line 254-259) “Galicía *et al.* reported that the K14, R115, K120 residues in MglB are highly conserved in MglB homologs, surface-exposed in the solved structure of the

MgIB dimer generating two positively charged surface regions on the four-helix side of the MgIB dimer⁴⁰ (Fig. 3a). Moreover, they reported that the MgIB^{K14A R115A K120A} variant (henceforth, MgIB^{KRK}) with substitutions of these three positively charged residues to Ala still has GAP activity *in vitro* but localizes diffusely by an unknown mechanism⁴⁰. Next, we describe why we picked this MgIB variant for further analysis (line 259-262) “Because MgIB localizes diffusely in the absence of MglC, we, therefore, hypothesized that the positively charged surface regions in the MgIB dimer defined by the K14, R115, K120 residues could be involved in the interaction between MgIB and MglC”.

Our *in vitro* experiments demonstrate (line 263-264) “His₆-MgIB^{KRK} did not detectably bind Strep-MglC” and our *in vivo* experiments demonstrate (line 264-266) that “polar localization of MgIB^{KRK}-mCherry in otherwise WT cells was strongly reduced independently of the presence or absence of MglC and MglA”. Conversely, the MgIB^{KRK} variant caused reduced – but not abolished – polar localization of MglC, RomR and MglA (line 266-268). Altogether, we therefore conclude that (line 268-270) “MgIB^{KRK} is deficient in interacting with MglC and suggest that the positively charged KRK surface regions in the MgIB dimer represent the interface to MglC”. This conclusion is also in agreement with a previous proposal by Kapoor et al.⁴³. Finally, in our AlphaFold model of the MglC₂(MgIB₂)₂ complex (line 362-371), we find that the MgIB R115 residue (from KRK) is in close proximity to the FDI residues of MglC, that are important for the MglC/MgIB interaction, reinforcing the idea that the KRK surface region in MgIB interfaces with the FDI surface region in MglC. To emphasize this point, we redid Fig. 6a to illustrate the close proximity of R115 to the FDI region.

Concerning the comment “What is the explanation for how these mutations could result in a complete disruption of the complex? Since this MgIB mutant does not affect MglA GAP activity (Galicia et al, 2019), why should it affect MglA-mVenus localization (line 263)? If the MglA localization is affected, it should imply that the interaction between MglA and MgIB is affected?”: Again we apologize for not being clearer. The explanation lies in the many interactions between these proteins. To clarify, we have now included in the main text (line 271-274) “Moreover, we suggest that the effect of the MgIB^{KRK} variant on RomR and MglA localization is caused by the interruption of the RomR/MglC/MgIB positive feedback, resulting in reduced polar RomR/RomX localization and, consequently, reduced MglA polar localization”.

b) MglC-FDI mutant

Line 275-276: Given the difficulties faced in purification of MglCFDI complex, how have the authors ensured that the corresponding mutation results in a folded and functional protein within the Myxococcus cells? Could it be that the effect observed was due to the partial unfolding of MglC resulting in a proportion of non-functional protein?

All the observed phenotypes of the mutant (lines 275 - 283) are equivalent to those of a deletion of MglC? Also in Supp Fig. 5, the amount of MglC-FDI-mVenus is quite weak compared to that of MglC-WT-mVenus.

Response: Thanks & great point! We followed the advice of the reviewer and included in the main text that the MglC^{FDI} and MglC^{KRR} for inclusion bodies when overexpressed in *E. coli* (line 286, 311). Moreover, we included new experiments in which we tested the MglC^{FDI}-mVenus and MglC^{KRR}-mVenus for solubility in *M. xanthus*. We find that both proteins are soluble. These new experiments are described in line 287-288 and 312-313 and included in Supplementary Fig. 5b. We would also like to add that both mVenus-tagged proteins are fluorescent in *M. xanthus*

demonstrating that the tag is folded. Moreover, MglC^{FDI}-mVenus is polarly localized in a manner that (just like the WT variant) depends on RomR.

The reviewer is correct that MglC^{FDI}-mVenus accumulates at a reduced level. In line 288-289, we now include specific mentioning of this observation.

Regarding the comment “All the observed phenotypes of the mutant (lines 275 - 283) are equivalent to those of a deletion of MglC”: all the phenotypes we observe are indeed equivalent to those of a deletion of mglC. We rewrote to emphasize this point (line 297-301) “Moreover, we suggest that the effect of the MglC^{FDI} variant on RomR and MglA localization is caused by the interruption of the RomR/MglC/MglB positive feedback, resulting in reduced polar RomR/RomX localization and, consequently, reduced MglA polar localization, as observed in the $\Delta mglC$ mutant”.

c) MglC-KRK mutant:

Same question as above is relevant for MglC-KRR as well. Are these signatures of unfolded non-functional protein rather than specific effects of the mutation?

Response: Please see previous comment concerning the solubility of the MglC^{KRR}-mVenus variant.

Though I do understand that it is difficult to confirm the folded state of the mutants in vivo, the authors have to be cautious in interpreting the role of these residues. This can only be a speculation, but not a strong proof. Can an MglB-R115 or MglC-D mutant instead of the triple mutants be purified and tested in vitro and in vivo? This might provide the experimental proof for the hypotheses proposed by the authors.

Response: We very much appreciate this comment. We have carefully checked how we describe these mutants/variants. In the entire text, we take great care to state that the mutants/variants support/suggest a specific interaction. For instance, for the MglB^{KRK} variant, we write (line 268-374) “We conclude that MglB^{KRK} is deficient in interacting with MglC and suggest that the positively charged KRK surface regions in the MglB dimer represent the interface to MglC, in agreement with the suggestion by Kapoor *et al.*⁴³. Moreover, we suggest that the effect of the MglB^{KRK} variant on RomR and MglA localization is caused by the interruption of the RomR/MglC/MglB positive feedback, resulting in reduced polar RomR/RomX localization and, consequently, reduced MglA polar localization”.

In case of the AlphaFold models, we also take great care to write that these models support/suggest a particular interaction. For instance, for the MglC₂:(MglB₂)₂ model, we write (line 269-371) “ Thus, this structural model agrees with a 2:4 stoichiometry of the MglC/MglB complex and together with our experimental findings support that the oppositely charged MglB KRK and MglC FDI surface regions interface”. Finally, we would like to add that the major take home message from the manuscript is the RomR/MglC/MglB positive feedback and the MglA negative feedback. These conclusions are valid independently of the precise mode of interaction between the involved proteins. Therefore, we have not generated the any of the possible nine MglB and MglC variants with single amino acid substitutions.

2. RomR oligomerization data:

Lines 304 - 306: Will the complementary construct (369 - 420 of RomR) form a stable trimer? Have the authors checked this possibility?

Response: Once again: Great point & thanks! To provide positive evidence that RomR^C (=369-420 of RomR) alone is sufficient to form dimers and trimers, interact with MglC and localize polarly in *M. xanthus*, we included new experiments in which we analyze RomR^C in more details. Specifically, we generated a MalE-RomR^C variant for *in vitro* analyses and find (line 333-334; Fig. 5b) that MalE-RomR^C in mass photometry experiments was also detected with masses matching well monomers, dimers and trimers. Moreover, we find in pull-down experiments that MalE-RomR^C interacts with Strep-MglC (line 343-344; Fig. 5c). Finally, as described in line 349-351, we expressed the RomR^C-mCherry variant in *M. xanthus*. Even when this variant was expressed from the strong *pilA* promoter, it accumulated at a strongly reduced level (Supplementary Fig. 5f); nevertheless, most cells had a weak polar signal (Fig. 5f). Because the RomR^C-mCherry variant accumulated at a strongly reduced level, we did not test whether RomR^C is sufficient to recruit MglC and MglB to the poles. Altogether, these new experiments (together with the experiments with the RomR¹⁻³⁶⁸ variants) allow us to conclude (line 352-354) that the negatively charged RomR^C has three functions: It is required and sufficient for RomR oligomerization, represents the RomR interface to MglC, and is required and sufficient for the polar localization of RomR.

The impurities in the RomR purification makes the MP data inconclusive and the trimer formation shaky. Use of SEC-MALS coupled with checking the eluted fractions on SDS-PAGE will be a more reliable method for confirming the trimer formation (and complex formation as well - refer next point) in the absence of a pure preparation of MalE-RomR protein sample.

Response: All the impurities in the MalE-RomR, MalE-RomR¹⁻³⁶⁸ and MalE-RomR^C purifications are degradation products. We have tried multiple different procedures for purification of the MalE-RomR variants to minimize the degradation problem, e.g. different temperatures, addition of high concentrations of protease inhibitors and using the proteins immediately after purification. But have not been able to solve this problem. Importantly, we believe that these smaller proteolytic fragments do not alter our conclusions. Specifically, the MP data with MalE-RomR fits nicely with dimers and trimers, and the truncated version of MalE-RomR¹⁻³⁶⁸ is clearly a monomer based on MP. The fact that we see a trimer for full-length RomR is, therefore, strong evidence that the full-length protein makes a trimer. This is confirmed by our new MP experiments with MalE-RomR^C, which we also detect with masses matching well monomers, dimers and trimers (Fig. 5b). Similarly, the MP data with MglC and the RomR variants (Fig. 6b), are consistent with a RomR:MglC stoichiometry of 2:2 and 3:2 and a RomR^C:MglC stoichiometry of 2:2 and 3:2. Therefore, we have not done SEC-MALS experiments. Finally, we have amended all figure legends with MalE-RomR variants to include that the faster migrating proteins are proteolytic degradation products.

The quantitative immunoblot analysis data: please include the reference to a citation or an appropriate figure. I could not find where the data has been presented.

Response: We apologize. These data are included in Supplementary Fig. 5e.

Line 434 - 435 - Couldn't find the data for this observation. Please clarify and cite the relevant figure number. Correct Rom-C to RomR-C

Response: We apologize. We clarified the text (now line 495-497) to "*In vivo* quantification of the RomR concentration (Supplementary Fig. 5e) suggests that trimeric RomR is the predominant form *in vivo*; however, which of the two forms represent the active form remains to be determined". And, we corrected Rom-C to RomR^C. Thanks.

3. RomR-MgIC interaction:

The higher molecular weights in MP can be interpreted in multiple possibilities (for example, 297 could be RomR trimer plus MgIC monomer given the surprisingly high level of monomeric MgIC, which is expected to be an obligate dimer similar to MgIB). In the absence of assays similar to gel filtration, which will demonstrate a shift as well as the eluted fractions can be confirmed on an SDS-PAGE gel to establish presence of the complex, the data from MP is not convincing. Again, can this be established using a RomR 369 - 420 construct?

Also, the stoichiometry is not very compatible with the symmetry of MgIC, where the proposed interacting residues are at the symmetric dimeric interface, and it is difficult to imagine how a trimer of RomR could fit in here.

Response: The MP data with MalE-RomR fits nicely with dimers and trimers, and the truncated version of MalE-RomR¹⁻³⁶⁸ is clearly a monomer based on MP. The fact that we see a trimer for full-length RomR is, therefore, strong evidence that the full-length protein makes a trimer. This is confirmed by our new MP results with MalE-RomR^C, which we also detect with masses matching well monomers, dimers and trimers (Fig. 5b). Similarly, the MP data with MgIC and the RomR variants (Fig. 6b), are consistent with a RomR:MgIC stoichiometry of 2:2 and 3:2 and a RomR^C:MgIC stoichiometry of 2:2 and 3:2.

All our data support that (1) MgIC can interact with MgIB and RomR in parallel, and (2) a dimer or trimer of RomR interacts via its negatively charged dimeric or trimeric RomR^C with the positively charged KRR regions on the four-helix side of the MgIC dimer. However, as mentioned in the text (line 383-386) "To obtain structural insights into the RomR/MgIC complexes, we attempted to generate AlphaFold-Multimer structural models of dimeric and trimeric RomR, dimeric and trimeric RomR^C, RomR₂:MgIC₂ and RomR₃:MgIC₂ as well as RomR^C₂:MgIC₂ and RomR^C₃:MgIC₂ complexes. However, none of these complexes was predicted with high confidence". So, we clearly do not have detailed structural insights into the details of the RomR/MgIC interaction. Of note, we do not believe that these structural details are required to bring across the take-home message of this manuscript. It is also clear from the all the different AlphaFold-Multimer structural models that we have generated that there is something odd about RomR^C that makes these structural models low confidence. To circumvent this problem, we are currently trying to obtain the structure of RomR^C₂:MgIC₂ and RomR^C₃:MgIC₂ complexes.

4. Allosteric model based on AlphaFold prediction:

Line 424-426: This allosteric mechanism is questionable. AlphaFold does not provide a proof for it, but only a weak speculation. The dimeric MgIB structure generated from AlphaFold also matches with that of the MgIB/MgIC structure with similar helix orientations, whereas the experimental MgIB dimer structure (6HJM) resembles that of MgIAB complex (6IZW). Hence, it is possible that the differences in orientation is an artefact of AlphaFold prediction, and may or may not reflect an allostery based mechanism.

Response: As a starting point, we would like to mention that whenever we refer to the AlphaFold models, we specifically write that these models support/suggest a particular interaction. We never state anywhere that an AlphaFold model provides proof of something.

Concerning the different MgIB structures: In the revised manuscript, we rechecked all our AlphaFold models and we have taken great care to compare the different MgIB structures.

Firstly, the AlphaFold model of MglB₂ alone and the solved structure of MglB₂ (pdb ID: 6hjm⁴⁰) agree well with each other (line 358-359 supplementary Fig. 7b).

Secondly, we included in the revised manuscript (line 371-373; new Supplementary Fig. 7e) that MglB₂ bound to MglC₂ is structurally different compared to MglB₂ in isolation, i.e. the four-helix side is in a more closed state when complexed with MglC₂.

Thirdly, as described (line 435-438), when comparing the solved, crystallographic structure of MglA-GTPγS:MglB₂ (pdb ID: 6izw⁴¹) to the MglC₂:(MglB₂)₂ AlphaFold-Multimer model (Fig. 6a), we identified significant conformational differences in the MglB dimers in the two complexes. Specifically, the MglB₂ four-helix side is in a more open state when complexed with MglA-GTPγS than with MglC₂ (Fig. 7c). As a result, the two R115 residues in the MglB KRK regions are likely positioned in such a way in the complex with MglA-GTPγS that they cannot interact with D26 and I28 in the MglC FDI region (Fig. 7c,d; see also Fig. 6a).

Fourthly, we added (line 438-440; new supplementary Fig. 8b, c) that “We note that in the solved structure of the MglB dimer with MglA-GTPγS, the four-helix side is more open than in the solved structure of the MglB dimer in isolation (Supplementary Fig. 8b). Thus, these solved structures together with the AlphaFold-Multimer model of the MglC₂:(MglB₂)₂ complex support that the MglB dimer can exist in three different conformational states, where the degree of “openness” on the four-helix side varies (Supplementary Fig. 8c)”. Based on these comparisons, we do not agree with the reviewer that “dimeric MglB structure generated from AlphaFold also matches with that of the MglB/MglC structure” and that “experimental MglB dimer structure (6HJM) resembles that of MglAB complex (6IZW)”.

Superposition of MglB dimer structure with 6IZW is an overall superposition of the dimer. The best demonstration of a relative movement between the two monomers of MglB comes up when one of the protomers is superposed. Such a superposition shows that there is an angular difference between the two protomers in the AlphaFold generated MglB dimer, compared to the structures of MglB with MglA (6IZW) or MglB alone (6HJM). Redo the figure panel in the Supplementary accordingly.

Response: Please see our response to the previous comment.

5. Positive feedback loop:

The positive feedback loop assumes that RomR recruits C and then B. However, a mechanism based on breaking the interface between MglC and MglB implies that the levels of MglB will not be commensurate with that of RomR, both of which are localised at the lagging pole. Won't such a model lead to a loss of MglB from the lagging pole too, and not only detachment of MglA? How can a RomR/MglC/MglB positive feedback function in this case?

Response: Good point! In the revised manuscript, we included a discussion of what happens at the lagging cell pole (line 485-491).

Can the FRAP experiments be extended to labeled MglC and MglB also to provide proof for the positive feedback?

Response: We agree that extending the FRAP experiments to MglC and MglB could provide further insights into the system. In previous experiments, polar MglB was shown in FRAP experiments to exchange dynamically with the cytoplasm with a $T_{1/2}$ of ~6sec³⁴. Because this exchange is very fast and technically challenging to capture, we decided not to pursue FRAP experiments with MglB and also not with MglC. We do not believe this information is essential for this manuscript as our fluorescence microscopy data together with the FRAP data we

present for RomR, clearly demonstrate that RomR, MglC and MglB reinforce each other's polar localization.

In panel 7A (2nd gel), can the observation be explained by the fact that MglB has been sequestered by MglA? Has the complex formation of MglAB been monitored by an equivalent experiment with a tag on MglA or MglB?

Response: Concerning the first question: The answer is yes, this is exactly what we think is happening. We have amended the text (line 427-431) to "We conclude that MglA-GTP specifically breaks the MglC/MglB interaction but not the MglC/RomR interaction. This observation is also in agreement with neither MglC nor RomR/MglC affecting MglB and MglB/RomY GAP activity (Supplementary Fig. 4). Thus, addition of MglA-GTP to the RomR/MglC/MglB complex results in sequestration of MglB by MglA-GTP, thus allowing MglB GAP activity to proceed".

Concerning the second question: In the experiments in Fig. 7a, we pulled on MglC because that allowed us to distinguish whether MglA breaks the MglB/MglC or the MglC/RomR interaction. Because MglB as well as MglA are His6-tagged, an experiment in which we pull on MglB is not technically possible.

Similarly, what is the explanation of minor binding in the 3rd panel? Here too, MglA is capable of binding to MglB with GDP as well (Baranwal, et al, 2019).

Response: We specifically write (line 423-425) "Intriguingly, in the presence of MglA-His₆ loaded with GppNHp, Strep-MglC no longer pulled-down His₆-MglB but still pulled-down His₆-MglB in the presence of MglA-His₆ loaded with GDP (Fig. 7a)". Because we do not comment quantitatively on the minor effect of MglA-GDP on the MglC/MglB interaction, we would prefer to keep the text as is.

6. GAP and GEF activities at the lagging pole:

Lines 467 - 469: Is it possible that the RomRX GEF activity is not active or relevant at the lagging pole, but is required only at its low levels in the leading pole? The idea of both GAP and GEF being necessary at the lagging pole is not clear. The system should be an efficient GAP and an inefficient GEF at the lagging pole for MglA localization to be correctly established?

Response: Overall, we agree with the reviewer. In a simple scenario, *M. xanthus* would have a GEF at the leading and a GAP at the lagging pole. This configuration would allow cells to move unidirectionally. However, *M. xanthus* cells not only move unidirectionally, they also reverse their direction of motility. Such reversals depend on an inversion of the polarity of the proteins of the polarity module. And, it is precisely these polarity inversion events that necessitate the paradoxical/peculiar colocalization of the GEF and GAP complexes at the lagging pole:

As mentioned in the Introduction and in the Discussion, it is a paradox that the GEF and GAP colocalize with their highest concentrations at the lagging pole. As described in details in the Introduction (line 93-108), high concentrations of polar MglB stimulate polar recruitment of its low-affinity interaction partner RomY²⁸. Because the MglB/RomY complex has higher GAP activity than MglB alone, the GAP activity dominates over the GEF activity at the lagging pole²⁸. Consequently, MglA-GTP recruitment to this pole is prohibited. At the opposite pole, the RomR/RomX GEF activity dominates over GAP activity because the low concentration of MglB is insufficient to recruit RomY²⁸. Consequently, MglA-GTP is recruited to this pole. In the current manuscript, we show that the paradoxical GEF complex at the lagging pole is important for

reversals. Therefore, we write in the Discussion (line 523-533) “We, therefore, suggest that the spatial configuration of the polarity proteins in the $\Delta mgIC$ mutant makes it less sensitive to Frz signaling because there is too little RomR/RomX GEF at the lagging pole to recruit MglA-GTP during reversals. Thus, the RomR/MglC/MglB positive feedback resulting in the peculiar colocalization of the RomR/RomX GEF and MglB/RomY GAP at the lagging pole in WT serves two purposes: First, the GAP activity displaces MglA-GTP from this pole to enable unidirectional translocation; and, second, the GEF activity is necessary to provide the system with the ability to rapidly and efficiently invert polarity. In other words, an important role of MglC and the RomR/MglC/MglB positive feedback is to establish the configuration of the polarity proteins that confer the polarity module with responsiveness to the Frz system”.

Supp Fig 4: According to the allosteric mechanism, the presence of all the components should either break the interface between MglA and MglB and hence reduce the GAP activity? How can the authors claim high GAP activity to drive the MglA from the lagging pole, given that the allosteric model is valid?

Response: MglA-GTP specifically breaks the MglC/MglB interaction but not the MglC/RomR interaction. Accordingly, in the GTPase assay neither MglC nor RomR/MglC affect MglB and MglB/RomY GAP activity (Supplementary Fig. 4). To more clearly point out how these two observations are consistent, we have amended the text to write (line 427-431) “We conclude that MglA-GTP specifically breaks the MglC/MglB interaction but not the MglC/RomR interaction. This observation is in agreement with neither MglC nor RomR/MglC affecting MglB and MglB/RomY GAP activity (Supplementary Fig. 4). Thus, addition of MglA-GTP to the RomR/MglC/MglB complex results in sequestration of MglB by MglA-GTP, thus allowing MglB GAP activity to proceed” and (line 485-491) “In our model, the high concentration of the MglB/RomY GAP complex at the lagging pole inhibits MglA-GTP recruitment to this pole by stimulating MglA GTPase activity. In this process, MglA-GTP breaks the RomR/MglC/MglB positive feedback resulting in detachment of MglB. However, due to the high concentration of RomR and MglC at this pole, MglB will rapidly be recaptured to restore the RomR/MglC/MglB complex. Consistent with this notion, polar MglB exchanges rapidly with the cytoplasm with a $T_{1/2}$ of ~6sec in FRAP experiments³⁴”.

Line 440: It is proposed in Szadkowski et al, 2022, that RomY interacts with MglB and an MglA-GTP state. Since MglA-GTP-MglB and MglB-MglC are mutually exclusive, is it correct propose a RomR/MglC/MglB.RomY complex?

Response: In Szadkowski et al. (Ref 28) we show that RomY interacts with MglB alone as well as with MglA-GTP alone with low affinity. In vivo, RomY specifically localizes to the lagging pole with the high concentration of MglB. So, we think it is correct to propose the RomR/MglC/MglB/RomY complex.

Minor points:

1. Lines 174 - 178: How is it concluded that the localization strongly depends on MglC? This conclusion can be made only in a background where only MglC is deleted? In delta MglC, MglB-mCherry was strongly reduced (line 151).

Response: In Fig. 1c, we show that MglB localization is strongly reduced in the absence of MglC. In the experiments described in line 173-177, we show that MglB localization is also strongly reduced in the $\Delta mglA\Delta mgIC$, the $\Delta mglA\Delta romR$ and the $\Delta mglA\Delta mgIC\Delta romR$ mutants (Fig. 1f). These results confirm that MglB-mCherry polar localization depends strongly on MglC

and, as previously shown¹⁷, on RomR. Moreover, neither MglC nor RomR alone can establish efficient polar MglB-mCherry localization.

2. Line 181-182: The basis of this conclusion that ‘neither MglC or MglB alone stimulates polar localization of RomR’ is not convincing given the decrease in RomR upon deletion of MglB or MglC.

Response: Thanks for pointing this out to us. We rewrote the sentence to (line 176-177) “Moreover, only when MglB and MglC are both present can they further stimulate RomR polar localization”.

3. Line 184: How is the order of recruitment derived based on the observations? It need not be the only option.

Response: We have indeed played around with all different possibilities. However, this is the only one that fits with the experimental data. Moreover, the in vivo data very nicely fits with the in vitro pulldown experiments.

4. The final R for RomR is missing in a couple of instances in the Discussion.

Response: Thanks & corrected throughout.

Reviewer #3 (Remarks to the Author):

Regulated reversals of the direction of individual cell movement enable populations of *Myxococcus xanthus* bacteria to coordinate multicellular behaviors such as predatory feeding and starvation-induced fruiting body development. Elucidating the regulation of cell reversal is key to understanding the multicellular behaviors of *M. xanthus* and, more broadly, the mechanisms by which cells dynamically control polar localization of proteins. Many proteins involved in *M. xanthus* polarity control have been described, but important mechanistic and design questions have remained. The authors report that MglC interacts with RomR and MglB at the rear pole of the cell, where GAP activity of MglB/RomY inhibits MglA-GTP accumulation and assembly of motility machineries. They also provide evidence that MglA-GTP at the front pole inhibits the interaction of MglB with MglC. Although RomR/RomX GEF activity at the rear pole would tend to stimulate MglA-GTP accumulation, thus counteracting MglB/RomY inhibition, this design appears to poise the system for rapid reversal upon signaling from the Frz chemosensory system. The authors compare this design with two others involving GTPase activity; yeast polar budding with a simpler design, and mammalian cell growth regulation with comparable complexity to *M. xanthus* in terms of GTP/GDP control of protein-protein interactions.

The work significantly advances understanding of the cell polarity regulatory network in *M. xanthus*. Previously, the Sogaard-Andersen group identified MglC as a paralog of MglB and used phenotypic and genetic approaches to begin characterizing MglC function (McLoon et al. 2016). Here, the authors use a battery of approaches to take the work much further by quantifying polar localization of functional fluorescent fusions to MglC, MglB, RomR, and MglA in a large number of single, double, and triple knockout mutants, purifying tagged proteins and performing biochemical pull-down assays (including with protein variants to identify interacting surfaces), characterizing oligomerization and functions of the RomR C-terminal domain, using FRAP to show that MglC and MglB stabilize RomR polar localization, using pull-downs to show

that MglA-GTP breaks the MglC/MglB interaction but not the MglC/RomR interaction, and proposing a model of the regulatory interactions at the front and rear poles.

The work supports the conclusions and in general is very well done. There is room for some improvements (see comments below), but this should not prohibit publication. The methodology is sound and well-described, and in general the manuscript is very well-written for a broad audience.

Response: Thank you very much for the very positive, helpful & constructive feedback.

Comments:

The Abstract (l. 31-34) is difficult to understand because the positive and negative feedback are not adequately explained. The authors state “that RomR and the MglB and MglC roadblock domain proteins generate a positive feedback by forming a RomR/MglC/MglB complex”, but this fails to explain to the nature of the positive feedback. It seems better to say “that RomR and the MglB and MglC roadblock domain proteins enhance each other’s polar localization by forming a RomR/MglC/MglB complex, ...”. Likewise, the following sentence would be easier to understand as “MglA at the front inhibits formation of the RomR/MglC/MglB complex by binding to MglB and breaking its interaction with MglC, ...”. The positive and negative feedback could then be explained in the Introduction if the authors want to use those terms.

Response: Thank you very much for these great suggestions. We tried to implement them in the Abstract but because we are limited to 150 words, we in the end decided not to change the Abstract.

l. 84 – the authors state that “MglB alone has GAP activity” and they never mention that it has GEF activity, yet ref. 41 states that “The dual GAP-GEF activities of MglB accelerate the rate of GTP hydrolysis over multiple enzymatic cycles.” Why don’t the authors mention the GEF activity of MglB?

Response: Thanks for pointing this out to us. We decided not to include mentioning of this GEF activity because its in vivo relevance has not been documented.

l. 129 – a gliding motility defect of the mglC mutant is not evident from Fig. S1A since there appear to be as many single cells as for WT at the edge of the colony on 1.5% agar. However, I’m not sure what are single cells (arrows may help) even when viewed at 800%.

Response: We apologize and have included new experiments at lower and higher magnifications in Supplementary Fig. 1a to illustrate that the $\Delta mglC$ mutant spreads less than the WT under conditions optimal for gliding and still has single cells at the colony edge.

Fig. S1C – the frzE mutant appears to move faster than WT on 1.5% agar. Was this known? Why would it be the case?

Response: We also noticed this difference. As far as we know, this has not previously been published. In the context of this manuscript, we would prefer not to go into a discussion of this observation because we do not think that it is relevant for all the other results presented.

Fig. S1C - why is the speed and reversal not measured for the complemented mglC mutant and the MglC-mVenus strain?

Response: In the population-based motility assays in Supplementary Fig. 1a, we find that the complemented $\Delta mgIC$ mutant and the MglC-mVenus strain behave like WT. Therefore, we did not do the speed and reversal frequency experiments for these two strains.

Fig. 1C – in the panel showing MglA-mVenus, the slope of the gray lines is not -1, as in the other panels. Is there a mistake?

Response: Thank you very much for pointing this out to us. We have corrected all the figures with MglA-mVenus.

I. 275-278 – this part is weak. The authors state they were unable to purify a soluble Strep-tagged MglC FDI variant. Did the protein accumulate? Did the authors try to overcome this problem by changing only D to A, or only D to K (charge reversal), since F and I might contribute to the hydrophobic core such that A at both those positions leads to unfolding, and only D contributes to the negatively charged surface? The other weakness is that MglC FDI-mVenus accumulates poorly relative to MglC-mVenus (Fig. S5B), which could affect the polar localization of the variant. Did the authors try to increase MglC FDI-mVenus accumulation by fusing the gene to the *pilA* promoter, which appeared to increase the MglC FDI level in the *mgIC* mutant relative to the MglC level in WT (Fig. S5B)? If these issues cannot be resolved or the authors do not wish to do additional experiments, Fig. 4 could be moved to the Supplementary Information.

Response: Great point & thanks! We followed the advice of the reviewer and included in the main text that the MglC^{FDI} and MglC^{KRR} form inclusion bodies when overexpressed in *E. coli* (line 286, 311). Moreover, we included new experiments in which we tested the MglC^{FDI}-mVenus and MglC^{KRR}-mVenus for solubility in *M. xanthus*. We find that both proteins are soluble. These new experiments are described in line 287-288 and 312-313 and included in Supplementary Fig. 5b. We would like to add that both mVenus-tagged proteins are fluorescent in *M. xanthus* demonstrating that the tag is folded. Moreover, MglC^{FDI}-mVenus is polarly localized in a manner that (just like the WT variant) depends on RomR.

In the revised text, we now also more carefully describe that in the solved structure of the MglC dimer, the FDI residues are surface exposed (line 278-280). Because the untagged MglC^{FDI} variant accumulates as the WT protein and this strain behaves like the $\Delta mgIC$ mutant, we have not generated any of MglC variants with single amino acid substitutions.

Concerning the reduced accumulation of MglC^{FDI}-mVenus: For fluorescently-tagged proteins we always aim to express from the native site to reach an accumulation similar to that of the native protein. So, have not tried to expression with the *pilA* promoter.

I. 291-292 – as in my preceding comment, this part is weakened by inability to purify the MglC variant, which is surprising given that KRR are presumably surface-exposed residues. Similar questions as in the preceding comment arise. Did the protein accumulate? Did the authors try to overcome this problem by changing only one or two of the three charged residues? This part is not as weak as in the preceding comment, since in this case the MglC KRR-mVenus level was similar to that of MglC-mVenus (Fig. S5C), so the level of the variant should not affect polar localization. However, the *romR* single mutant was not tested for MglC KRR-mVenus localization (Fig. 4D). Granted, localization would very likely be lost since that was the case for MglC-mVenus (Fig. 1E), but it seems odd to leave it out since the authors infer that KRR of MglC interacts with RomR. In any case, the main difference between the FDI and KRR variants

of MglC is the effects of the mglB and mglA single mutants in panels B and D of Fig. 4, and I'm not convinced that this observation warrants two sections of Results and Fig. 4. The evidence for the inferred interaction surfaces of MglC with MglB and RomR is weak and could be moved to the Supplementary Information without detracting from the main conclusions of the manuscript.

Response: Please see the previous comment for the new experiments with the solubility test of MglC^{KRR}-mVenus.

In the revised text (line 313-316), we now clearly state that MglC^{KRR} with or without the mVenus tag accumulates at the WT protein. Because MglC^{FDI}-mVenus is soluble and accumulates well, we have not generated any of MglC variants with single amino acid substitutions. In Fig. 4c, we now also included data for the RomR mutant.

We believe that the analysis of the MglC variants provide important information, also in the context of the AlphaFold models. Therefore, we decided to keep these results in the main text.

Fig. 5C – the authors should offer an explanation for the multiple species below MalE-RomR 1-368, perhaps in the legend. Presumably these are proteolytic fragments of the truncated fusion protein.

Response: Good point! All the impurities in the MalE-RomR, MalE-RomR¹⁻³⁶⁸ and MalE-RomR^C purifications are degradation products. Therefore, we have amended all figure legends with MalE-RomR variants to include that the faster migrating proteins are proteolytic degradation products.

Fig. 7F top part – does not depict RomR associated with the cell envelope at the pole, yet RomR is crucial for polar localization of the other proteins and the authors state that polar RomR recruits MglC (l. 414). If the arrow from RomR to itself, touching the cell envelope, is meant to depict positive autoregulatory cell envelope association, this should be explained in the legend and the corresponding arrow at the leading pole should touch the cell envelope.

Response: Many apologies for not being clear. We have amended the legend to Fig. 7f to make clear what the arrow from RomR on itself means.

At the end of the Discussion, the authors very nicely come back to the yeast polar budding mentioned in the Introduction and compare it with their findings about *M. xanthus* polarity control. They also make an interesting comparison with mammalian cell growth regulation, which is comparable in complexity to *M. xanthus* motility regulation and likewise involves GTP/GDP control of protein-protein interactions. Perhaps due to a word limit, the authors did not compare their findings with other well-studied examples of regulation of bacterial cell polarity (e.g., *Caulobacter* swarmer versus stalk cell differentiation, *M. xanthus* cell division, etc.). The authors are expert in this area (e.g., Treuner-Lange and Sogaard-Andersen 2014 *J Cell Biol*, Schumacher and Sogaard-Andersen 2017 *Ann Rev Microbiol*), so further comparison of design principles (mentioned in the Title) would broaden interest and likely be preferable to the two sections of Results and Fig. 4 mentioned in my preceding comments, which could be moved to the Supplementary Information.

Response: Many thanks! There is definitely something fascinating to be said about the very different systems that regulate polarity in bacteria. However, a comparison of the different systems would require several lengthy descriptions and discussions. So, we do not believe that

this is the right place to make this comparison. Therefore, we would like to stick to the comparison to the yeast Cdc42 system and the Rag/Ragulator system.

Minor points:

I. 193 – add “(Fig. 1F)” after MglC

Response: Thanks & done.

Fig. 1H – typo in the y-axis label – should be “highest”

Response: Thanks & corrected.

Fig. S3 legend mentions black circles but they are missing

Response: Thanks & legend corrected.

Fig. 4C legend – “reduces” should be “reduced”

Response: Thanks & corrected.

I. 285 – “interphase” should be “interface”

Response: Thanks & corrected.

Fig. 5B legend – “fittet” should be “fitted”

Response: Thanks & corrected.

Fig. 7C legend – very confusing because not all the colors are explained and “(red/brown)” seems to describe the MglC2:MglB2 model in reverse order. The colors in the inset also need to be explained better.

Response: Many apologies & corrected.

I. 413 – “KRR” should be “KRK”

Response: Thanks & corrected.

I. 475 – “centred” should be “centered”

Response: Thanks & corrected.

REVIEWERS' COMMENTS

Reviewer #1 (Remarks to the Author):

The authors have answered all my previous questions.

Reviewer #2 (Remarks to the Author):

The authors have addressed the concerns raised in the earlier review satisfactorily.

Reviewer #3 (Remarks to the Author):

The authors have addressed the reviewers' comments to my satisfaction and the manuscript is improved. The new results in Supplementary Figure 5b show that the MglC FDI- and MglC KRR-mVenus fusion proteins are soluble in *M. xanthus*, strengthening the interpretation of the localization results shown in Figure 4. The new results in Figure 5bc show that the MalE-RomR C-terminal domain fusion protein is sufficient for dimer and trimer formation and interaction with Strep-MglC. The explanations of the suggested interactions between MglB and MglC, and how they change upon MglA-GTP binding, are also improved.

Comment:

The new result in Figure 5f shows very little polar localization of the RomR C-terminal domain-mCherry fusion protein, so the evidence that the RomR C-terminal domain is sufficient for polar localization is very weak. Therefore, the statements related to this point should be qualified (e.g., "partly responsible for" rather than "sufficient" on l. 354 and 494).